# Neurostimulation in People with Oropharyngeal Dysphagia: A Systematic Review and Meta-Analyses of Randomised Controlled Trials—Part I: Pharyngeal and Neuromuscular Electrical Stimulation

**DOI:** 10.3390/jcm11030776

**Published:** 2022-01-31

**Authors:** Renée Speyer, Anna-Liisa Sutt, Liza Bergström, Shaheen Hamdy, Bas Joris Heijnen, Lianne Remijn, Sarah Wilkes-Gillan, Reinie Cordier

**Affiliations:** 1Department Special Needs Education, Faculty of Educational Sciences, University of Oslo, 0318 Oslo, Norway; 2Curtin School of Allied Health, Faculty of Health Sciences, Curtin University, Perth, WA 6102, Australia; 3Department of Otorhinolaryngology and Head and Neck Surgery, Leiden University Medical Centre, 1233 ZA Leiden, The Netherlands; b.j.Heijnen@lumc.nl; 4Critical Care Research Group, The Prince Charles Hospital, Brisbane, QLD 4032, Australia; annaliisasp@gmail.com; 5School of Medicine, University of Queensland, Brisbane, QLD 4072, Australia; 6Remeo Stockholm, 128 64 Stockholm, Sweden; liza.bergstrom@regionstockholm.se; 7Speech Therapy Clinic, Danderyd University Hospital, 182 88 Stockholm, Sweden; 8GI Sciences, School of Medical Sciences, Faculty of Biology, Medicine and Health, University of Manchester, Manchester M13 9PL, UK; shaheen.hamdy@manchester.ac.uk; 9School of Allied Health, HAN University of Applied Sciences, 6525 EN Nijmegen, The Netherlands; lianne.remijn@han.nl; 10Discipline of Occupational Therapy, Sydney School of Health Sciences, Faculty of Medicine and Health, The University of Sydney, Sydney, NSW 2006, Australia; sarah.wilkes-gillan@sydney.edu.au; 11Department of Social Work, Education and Community Wellbeing, Faculty of Health & Life Sciences, Northumbria University, Newcastle upon Tyne NE7 7XA, UK; reinie.cordier@northumbria.ac.uk

**Keywords:** deglutition, swallowing disorders, RCT, intervention, neuromuscular electrical stimulation, pharyngeal electrical stimulation, PES, NMES

## Abstract

*Objective.* To assess the effects of neurostimulation (i.e., neuromuscular electrical stimulation (NMES) and pharyngeal electrical stimulation (PES)) in people with oropharyngeal dysphagia (OD). *Methods.* Systematic literature searches were conducted to retrieve randomised controlled trials in four electronic databases (CINAHL, Embase, PsycINFO, and PubMed). The methodological quality of included studies was assessed using the Revised Cochrane risk-of-bias tool for randomised trials (RoB 2). *Results.* In total, 42 studies reporting on peripheral neurostimulation were included: 30 studies on NMES, eight studies on PES, and four studies on combined neurostimulation interventions. When conducting meta analyses, significant, large and significant, moderate pre-post treatment effects were found for NMES (11 studies) and PES (five studies), respectively. Between-group analyses showed small effect sizes in favour of NMES, but no significant effects for PES. *Conclusions.* NMES may have more promising effects compared to PES. However, NMES studies showed high heterogeneity in protocols and experimental variables, the presence of potential moderators, and inconsistent reporting of methodology. Therefore, only conservative generalisations and interpretation of meta-analyses could be made. To facilitate comparisons of studies and determine intervention effects, there is a need for more randomised controlled trials with larger population sizes, and greater standardisation of protocols and guidelines for reporting.

## 1. Introduction

The aerodigestive tract facilitates the combined functions of breathing, vocalising, and swallowing. Any dysfunction in this system may lead to oropharyngeal dysphagia (OD) or swallowing problems [1]. OD can be the result of underlying diseases such as stroke or a progressive neurological disease (e.g., Parkinson’s disease, multiple sclerosis) or an adverse effect after head and neck oncological interventions (e.g., radiation or surgery) or intensive care treatment (e.g., intubation and tracheostomy). Prevalence estimates of OD have been reported to be as high as 50% in cerebral palsy [2], 80% in stroke and Parkinson’s disease, and over 90% in people with community-acquired pneumonia [3]. OD can have a severe impact on a person’s health as it may lead to dehydration, malnutrition, and even death. Research has identified inverse bidirectional relationships between decreased health-related quality of life and increased OD severity [4].

Traditional OD therapy may include physical interventions such as: bolus modification and management (e.g., adjusting the viscosity, volume, temperature and/or acidity of food and drinks); oromotor exercises; body and head postural adjustments; and swallow manoeuvres (e.g., manoeuvres to improve food propulsion into the pharynx and airway protection) [1]. Therapy may also include sensory stimulation, which involves applying techniques like thermal stimulation and chemical stimulation using natural agonists of polymodal sensory receptors (e.g., capsaicin, the spicy component of peppers) [5]. 

Another type of stimulation considered to be beneficial for promoting rehabilitation of swallowing dysfunction is acupuncture. This practice emerged from traditional Chinese medicine and exerts therapeutic effects by inserting thin needles at strategic places, termed acupuncture points, on the body surface aiming to rebalance the flow of energy or life force (‘qi’). Needles are then activated through specific manual movements or electrical stimulation. Although stimulation of acupuncture points seems to be associated with places where nerves, muscles, and connective tissues may be stimulated [6], their intrinsic mechanisms are still part of a continuing scientific debate on acupuncture.

Recently, an increasing number of studies have been published on alternative interventions aiming to enhance neural plasticity by using non-invasive brain stimulation (NIBS) techniques. Repetitive transcranial magnetic stimulation (rTMS) and transcranial direct current stimulation (tDCS) are cortically or centrally applied NIBS techniques. Using electromagnetic induction, rTMS results in depolarisation of post-synaptic connections, whereas tDCS uses direct electrical current to shift the polarity of nerve cells [7]. Alternatively, electrical stimulation techniques like pharyngeal electrical stimulation (PES) and neuromuscular electrical stimulation (NMES) target the peripheral neural pathways [8]. NMES aims to strengthen muscular contractions during swallowing and uses stimulation by electrodes placed on the skin over the anterior neck muscles to activate sensory pathways [9,10,11]. In contrast, PES has been shown to drive neuroplasticity in the pharyngeal motor cortex through direct stimulation of the pharyngeal mucosa via intraluminal catheters [7].

Over the past decade, several reviews have been published on the effects of neurostimulation in patients with OD. Most of these reviews focused on selected types of neurostimulation: NMES [10,12], rTMS [13,14], tDCS [15], or rTMS and tDCS [16,17]. Only two systematic reviews included both cortical (rTMS and tDCS) and peripheral neurostimulation (PES and NMES) [18,19]. All reviews targeted interventions in post-stroke populations except one review that broadened inclusion criteria to patients with acquired brain injury including stroke [16]. To date, all systematic reviews on neurostimulation as a treatment for OD set boundaries for inclusion based on medical diagnoses. 

The aim of this systematic review is to determine the effects of neurostimulation in people with OD without excluding populations based on medical diagnoses. Findings are based on the highest level of evidence only, namely randomised controlled trials (RCTs), and summarised by conducting meta-analyses. The results of this review will be presented in two companion papers. This paper (Part I) reports on pharyngeal and neuromuscular electrical stimulation (PES and NMES) while the second paper (Part II) will report on brain stimulation (i.e., rTMS and tDCS).

## 2. Methods

The methodology and reporting of this systematic review were based on the Preferred Reporting Items for Systematic Reviews and Meta-Analyses (PRISMA) 2020 statement and checklist (Appendix A) which aim to enhance the essential and transparent reporting of systematic reviews [20,21]. The protocol for this review was registered at PROSPERO, the international prospective register of systematic reviews (registration number: CRD42020179842). 

### 2.1. Information Sources and Search Strategies

Literature searches to identify studies were conducted on 6 March 2021, across four databases: CINAHL, Embase, PsycINFO, and PubMed. Publication dates of coverage ranged from 1937–2021, 1902–2021, 1887–2021, and 1809–2021, respectively. Additional searches, including checking the reference lists of eligible articles, were performed. Two main categories of terms were used in combination: (1) dysphagia and (2) randomised control trials. Search strategies were performed in all four electronic databases using subheadings (e.g., MeSH and Thesaurus terms) and free text terms. The full electronic search strategies for each database are reported in Table 1. To identify other literature beyond that found using these strategies, the reference lists of each eligible article were checked.

### 2.2. Inclusion and Exclusion Criteria

Studies were included in this systematic review if they met the following criteria: (1) participants had a diagnosis of oropharyngeal dysphagia; (2) the study included non-invasive neurostimulation interventions aimed at reducing swallowing or feeding problems; (3) the study included a control group or comparison intervention group; (4) participants were randomly assigned to one of the study arms or groups; and (5) the study was published in the English language.

Interventions such as non-electrical peripheral stimulation (e.g., air-puff or gustatory stimulation), pharmacological interventions and acupuncture, were considered out of the scope of this review, and thus were excluded. Invasive techniques and/or those that did not specifically target OD (i.e., deep-brain stimulation studies after neurosurgical implementation of a neurostimulator) were also excluded. Conference abstracts, doctoral theses, editorials, and reviews were excluded.

Finally, only studies reporting on peripheral neurostimulation (i.e., PES and NMES) were included in this review (Part I). Studies on brain neurostimulation (i.e., rTMS and tDCS) will be reported on in a companion paper (Part II).

## 3. Systematic Review

### 3.1. Methodological Quality and Risk of Bias

The methodological quality of the included studies was assessed using the Revised Cochrane risk-of-bias tool for randomised trials (RoB 2) [22]. The RoB 2 tool identifies five domains to consider when assessing where bias may have been introduced into a randomised trial: (1) bias arising from the randomisation process; (2) bias due to deviations from intended interventions; (3) bias due to missing outcome data; (4) bias in measurement of the outcome; and (5) bias in selection of the reported result. The RoB 2 gives a series of signalling questions for each domain whose answers give a judgement (i.e., “low risk of bias,” “some concerns,” or “high risk of bias”), which can be evaluated to determine a study’s overall risk of bias [22]. 

### 3.2. Data Collection Process

A data extraction form was created to extract data from the included studies under the following categories: participant diagnosis, inclusion and exclusion criteria, sample size, age, gender, intervention goal, intervention agent/delivery/dosage, outcome measures, and treatment outcome. 

### 3.3. Data, Items and Synthesis of Results

Titles and abstracts of included studies were screened for eligibility by two independent reviewers, after which the eligibility of selected original articles was assessed by these same two reviewers. If agreement could not be reached between the first two reviewers, a third reviewer was consulted to reach consensus. Two independent researchers also assessed the methodological study quality and, where necessary, consensus was reached with involvement of a third reviewer. As none of the reviewers have formal or informal affiliations with any of the authors of the included studies, no evident bias in article selection or methodological study quality rating was present.

Data points across all studies were extracted using comprehensive data extraction forms. Risk of bias per individual study was assessed using the RoB 2 tool [22]. Data were extrapolated and synthesized using the following categories: participant characteristics, inclusion criteria, intervention conditions, outcome measures and intervention outcomes. Effect sizes and significance of findings were the main summary measures for assessing treatment outcome. 

## 4. Meta-Analysis

*Data Analysis.* Data were extracted from each study to compare the effect sizes for the following: (1) pre-post outcome measures of OD and (2) mean difference between neurostimulation and comparison controls in outcome measures from pre- to post-intervention. Control groups may receive no treatment, sham stimulation and/or traditional dysphagia therapy (DT; e.g., bolus modification, oromotor exercises, body and head postural adjustments, and swallow manoeuvres). Only studies using instrumental assessment (e.g., videofluoroscopic swallow study (VFSS) or fiberoptic endoscopic evaluation of swallowing (FEES)) to confirm OD were included. 

Data collected using outcome measures based on visuoperceptual evaluation of instrumental assessment were preferred over clinical non-instrumental assessments. Oral intake measures were only included if no other clinical data were available, whereas screening tools and patient self-report measures were excluded from meta-analyses altogether. When selecting outcome measures for meta-analyses, reducing heterogeneity between studies was a priority. Consequently, measures other than the authors’ primary outcomes may have been preferred if these measures contributed to greater homogeneity. 

To compare effect sizes, group means, standard deviations, and sample sizes for pre- and post-measurements, data were entered into Comprehensive Meta-Analysis Version 3.3.070 [23]. If only non-parametric data were available (i.e., medians, interquartile ranges), data were converted into parametric data for meta-analytic purposes. Studies with multiple intervention groups were analysed separately for each experimental-control comparison. If studies included the same participants, only one study was included in the meta-analysis. For studies providing insufficient data for meta-analysis, authors were contacted by e-mail to request additional data.

Effect sizes were calculated in Comprehensive Meta-Analysis using a random-effects model since it was unlikely that studies would have similar true effects due to variations in sampling, participant characteristics, intervention approaches, and outcome measurements. Heterogeneity was estimated using the *Q* statistic to determine the spread of effect sizes about the mean and *I*^2^ was used to estimate the ratio of true variance to total variance. *I*^2^-values of less than 50%, 50% to 74%, and higher than 75% denote low, moderate, and high heterogeneity, respectively [24]. Effect sizes were generated using the Hedges’ *g* formula for standardized mean difference with a confidence interval of 95%. Effects sizes were interpreted using Cohen’s *d* convention as follows: *g* ≤ 0.2 as no or negligible effect; 0.2 < *g* ≤ 0.5 as small effect; 0.5 < *g* ≤ 0.8 as moderate effect; and *g* > 0.8 as large effect [25].

Forest plots of effect sizes for OD outcome scores were generated for PES and NMES separately: (1) pre-post neurostimulation and (2) neurostimulation interventions versus comparison groups. Subgroup analyses were used to explore effect sizes as a function of various moderators depending on neurostimulation type. For example, outcome measures, medical diagnoses, total treatment duration, total neurostimulation time, and stimulation characteristics (e.g., pulse duration, pulse rate, electrode configuration). To account for the possibility of spontaneous recovery during the intervention period, only between-subgroup meta-analyses were conducted using post-intervention data.

Comprehensive Data Analysis software was utilized to evaluate publication bias. The Begg and Muzumdar’s test [26] was used to calculate the rank correlation between the standardised effect size and the ranks of their variances. The Begg and Muzumdar test calculates both a tau and a two tailed *p* value, with values of close to zero indicating no correlation, while results closer to 1 suggest a correlation. Where asymmetry is the result of publication bias, high standard error values would correspond with larger effect sizes. Where larger effects correspond to low values, tau would be positive (with the inverse also being true). Conversely, when larger effects correspond to high values, tau would be negative.

Publication bias was also evaluated utilising a fail-safe N test. This measure addresses the question of how many omitted studies would be necessary to nullify the effect. It refers to the number of studies where the effect size was zero being included in the meta-analysis prior to the result becoming statistically insignificant [27]. When this value is comparably low, there may be reason to treat the results with caution. When the value is comparably high, however, it can be reasonably concluded that the treatment effect is not nil, although it may be increased due to the omission of some studies. 

## 5. Results

### 5.1. Study Selection

A total of 8059 studies were identified through subject heading and free text searches from the four databases: CINAHL (*n* = 239), Embase (*n* = 4550), PsycINFO (*n* = 231), and PubMed (*n* = 3039). Removing duplicate titles and abstracts (*n* = 1113) left a total of 6946 records. A total of 261 original articles were assessed at a full-text level, with articles grouped based on type of intervention. Four additional studies were found through reference checking of the included articles. At this stage, no studies were excluded based on type of intervention (e.g., behavioural intervention, neurostimulation). Of the reviewed 261 articles, 58 studies on neurostimulation were identified that satisfied the inclusion criteria. As this systematic review reports on PES and NMES interventions only, a final number of 42 studies reporting on peripheral neurostimulation were included in this review. Figure 1 presents the flow diagram of the reviewing process according to PRISMA.

### 5.2. Description of Studies

All included studies are described in detail within Table 2 and Table 3. Specifically, Table 2 presents data on study characteristics including methodological study quality, inclusion and exclusion criteria, and details on participant groups. The following information is provided for all study groups (control and intervention groups): medical diagnosis, sample size, age and gender. Table 3 reports on intervention goals of included studies, intervention components, outcome measures, intervention outcomes, as well as main conclusions.

*Peripheral Neurostimulation Interventions*. Across the 42 included studies, 30 studies reported on NMES and eight studies reported on PES. Four studies used another type of neurostimulation (i.e., rTMS) in addition to NMES or PES, either within the same group or different treatment groups.

*Participants* (Table 2). The 42 studies included a total of 2281 participants (mean 54.3; SD 39.1). The sample sizes ranged from the smallest sample of 16 participants [60,61] to the largest sample of 162 participants [36]. By intervention type, samples were characterized as follows: NMES total 1706, mean 56.9, SD 38.9, range 18–135; PES total 410, mean 51.3, SD 49.0, range 16–162; and combined neurostimulation total 165, mean 41.3, SD 19.3, range 18–64. The mean age of participants across all studies was 61.8 years (SD 15.3), with one study reporting age range only (65–93 years) [61]. Participant mean age across all studies ranged from 4.2 years [54] to 84.4 years [39]. The mean age of participants by intervention group was: NMES 60.9 years (SD 16.9), PES 64.7 years (SD 11.9), and combined neurostimulation 63.8 years (SD 6.4).

Across all studies, 61.0% (SD 13.5) of participants were male and one study did not report gender distribution [30]. Percentage of males by intervention group was NMES 62.6% (SD 14.0), PES 56.7% (SD 9.6), and other/combined 65.4% (SD 12.3). Most studies included stroke patients (*n* = 31), while three studies included mixed populations [28,41,43] and one study reported OD without further underlying medical diagnosis [39]. Other diagnoses by intervention group were: Parkinson’s disorder (*n* = 2)[32,46], cerebral palsy (*n* = 2) [50,54], and head and neck cancer (*n* = 2) [36,48] in NMES; and multiple sclerosis (*n* = 1) [63] in PES.

Across the 42 studies, VFSS was most frequently used to confirm participant’s diagnosis of OD (*n* = 31), whereas six studies used FEES [49,53,54,60,64,65]. Several of these studies combined instrumental assessment with either a screen (*n* = 2) [58,65] or clinical assessment (*n* = 6) [49,50,53,54,55,68]. One study used either clinical assessment or VFSS [50]. One study used a single screen [56], three studies used clinical assessment only [35,38,59], and one study used both [33]. The studies were conducted across 14 countries, with studies most frequently conducted in Korea (*n* = 11), China (*n* = 7), the UK (*n* = 7), Spain (*n* = 4), Italy (*n* = 2), Turkey (*n* = 2), and Germany (*n* = 2).

*Outcome Measures* (Table 2). Outcomes measures varied greatly across all studies included in the review, covering several domains within the area of OD. The Penetration Aspiration Score was the most reported outcome measure (PAS; 18 studies), followed by Functional Oral Intake Scale (FOIS; 12 studies), Functional Dysphagia Scale (FDS; 5 studies), Dysphagia Severity Rating Scale (DSRS; 5 studies), Swallowing Quality of Life questionnaire (SWAL-QOL; 4 studies), and Dysphagia Outcome and Severity Scale (DOSS; 3 studies).

*NMES Intervention* (*n* = 30: Table 2 and Table 3). In total, 22 studies included two study arms or groups, whereas eight studies included three groups [31,32,33,34,38,40,55,57]. All but five NMES studies [29,39,43,53,54] combined neurostimulation with simultaneous DT consisting of a wide range of behavioural interventions (e.g., head and body positioning, bolus modification, oromotor exercises, or swallow manoeuvres). Six studies included a NMES only group without DT [29,33,38,39,43,55], with five of these studies using NMES at motor stimulation level [29,33,38,43,55] and one study using NMES at sensory stimulation level [39]. An additional seven studies included a treatment arm with NMES at sensory stimulation level combined with DT [32,44,45,46,53,54,57]. All other participants in NMES groups received stimulation at motor level. Five studies compared different NMES electrode positions [28,34,40,41,42] and seven studies included a sham stimulation group [36,39,48,50,52,53,54]. 

Control groups included mostly sham NMES stimulation and/or DT. Only one study included a control group receiving neither DT nor NMES [30], and one study included usual care across different healthcare settings as the comparison group [51]. 

*PES Intervention* (*n* = 8: Table 2 and Table 3). All eight studies compared PES to a sham version of the treatment [58,59,60,61,62,63,64,65]. None of the studies included other treatment groups (e.g., DT) or control groups (e.g., usual care or no treatment). 

*Combined Neurostimulation Interventions* (*n* = 4: Table 2 and Table 3). Three studies in the combined intervention group compared three different treatments. Of these, one study compared PES, paired associative stimulation (PAS) and rTMS [68], a second study compared DT, rTMS combined with DT, and NMES combined with DT [67], and a third study compared rTMS, PES and capsaicin stimulation [66]. A fourth study combined NMES stimulation with sham rTMS or rTMS stimulating different hemispheres (ipsilesional, contralesional or bilateral) [69]. 

### 5.3. Risk of Bias Assessment and Methodological Quality

The tau values from the Begg and Mazumdar rank correlation were 0.101 (two-tailed *p* = 0.589) and < 0.000 (two-tailed *p* > 0.999) for NMES and PES, respectively. The NMES meta-analysis incorporates data from 16 studies, which yielded a *z*-value of 4.107 (two-tailed *p* < 0.001). The fail-safe N is 55 indicating 55 ‘null’ studies need to be located and included for the combined two-tailed *p*-value to exceed 0.050. Therefore, there would need to be 3.4 missing studies for every observed study for the effect to be nullified. The PES meta-analysis incorporates data from five studies yielding a *z*-value of 1.156 (two-tailed *p* < 0.248). Since the combined result is not statistically significant, the fail-safe N (which addresses the concern that the observed significance may be spurious) is not relevant. Both of these procedures (i.e., Begg and Mazumdar rank correlation and fail-safe N) indicate the absence of publication bias.

Figure 2 and Figure 3 present, respectively, the risk of bias summary per domain for all included studies combined and for individual studies. The majority of studies had low risk of bias with very few exceptions.

## 6. Meta-Analysis: Effects of Interventions

### 6.1. Neuromuscular Electrical Stimulation (NMES) Meta-Analysis

Eleven studies were included in the NMES meta-analysis [28,29,34,37,40,42,45,47,49,51,55], of which six studies included two or three different intervention groups [28,34,40,42,45,55]. A total of 20 studies were excluded from meta-analysis for the following reasons: in three studies, OD diagnosis was not confirmed by instrumental assessment (VFSS or FEES); five studies provided insufficient data for meta-analyses; and, twelve studies were excluded to reduce heterogeneity: six studies including subject populations with medical diagnoses other than stroke (i.e., children with cerebral palsy, head and neck cancer patients, patients with Parkinson’s disease, and elderly), five studies because of outcome measures (e.g., kinematic or biomechanical variables in VFS recordings), and one study using sensory NMES stimulation.

*Overall within-group analysis* (Figure 4). A significant, large pre-post intervention effect size was calculated using a random-effects model (*z*(17) = 6.477, *p <* 0.001, Hedges’ *g* = 1.272, and 95% CI = 0.887–1.657). Pre-post intervention effect sizes ranged from 0.000 to 3.826. In 13 of the 18 NMES intervention groups, effect sizes were large (Hedges’ *g* > 0.8), indicating that NMES accounted for a significant proportion of standardized mean difference for these studies. Between-study heterogeneity was significant (*Q*(17) = 106.7, and *p* < 0.001), with *I*^2^ showing that heterogeneity accounted for 84.1% of variation in effect sizes across studies.

*Overall between-group analysis* (Figure 5). A significant, small post-intervention between-group total effect size in favour of NMES was calculated using a random-effects model (*z*(8) = 2.589, *p* = 0.010, Hedges’ *g* = 0.433, and 95% CI = 0.105–0.760). Between-study heterogeneity was significant (*Q*(8) = 18.0, and *p* = 0.021), with *I*^2^ showing that heterogeneity accounted for 55.6% of variation in effect sizes across studies.

Between-subgroup analyses. Subgroup analyses (Table 4) were conducted to compare diagnostic groups. Treatment effects were highest (moderate) for stroke patients, while other groups showed no significant effect sizes. For all other subgroup analyses, only stroke patients were included to improve homogeneity between studies. Subgroup analyses between studies compared intervention types (NMES, NMES + DT), time between pre- and post-intervention measurement, outcome measures, total stimulation times, electrodes configurations, pulse durations, and pulse rates (Table 4). NMES as an adjunctive treatment to DT showed significant, moderate positive treatment effects, whereas NMES alone showed non-significant effects. Effect sizes comparing time between pre- and post-treatment measurements showed no clear results. Although no effects could be identified at 2 weeks, a significant, positive effect size was found at 7 weeks. When comparing effect sizes based on outcome measures, the only significant effect found was a significant, large effect size for oral intake. The non-significant effects sizes for visuoperceptual evaluation of instrumental assessment ranged between negligible negative to moderate positive effects. Total stimulation time subgroup analyses showed significant, moderate positive treatment effects for longer stimulation times (>100 min). Shorter stimulation times did not result in significant effects. Comparisons for electrode configurations showed significant, moderate positive effects sizes for infrahyoid configuration. Electrode configuration based on patients’ characteristics, including OD outcome scores, indicated non-significant moderate effects, whereas both suprahyoid combined with infrahyoid and suprahyoid configurations resulted in negligible effects. Final comparisons between studies using different pulse durations did not suggest a linear relationship, whereas pulse rate comparisons indicated that studies using higher frequencies showed increased significant, positive moderate effect sizes. 

### 6.2. Pharyngeal Electrical Stimulation (PES) Meta-Analysis

Five studies using PAS in adult stroke patients were included in the meta-analyses [58,62,65,66,68]. Three studies were excluded from meta-analyses for the following reasons: overlap in participant population between studies, insufficient data for meta-analyses, and no confirmation of OD diagnosis prior to treatment.

Overall within-group analysis. The pre-post intervention effect sizes for the included studies ranged from 0.265 (small effect) [66] to 0.802 (large effect) [62], with an overall moderate effect size of 0.527 (Figure 6). As one study, however, did not provide PAS data for all included participants [65], a sensitivity analysis was conducted for both PAS and DSRS, indicating minimal differences in effect sizes. 

Overall between-group analysis. A non-significant post-intervention between-group total effect size in favour of PES was found using a random-effects model (*z*(4) = 0.718, *p* = 0.473, Hedges’ *g* = 0.099, and 95% CI = −0.170–0.368), suggesting no improvement in PAS outcomes following PES neurostimulation (Figure 7). Between-study heterogeneity was non-significant (*Q*(4) = 1.8, and *p* = 0.766). 

Between-subgroup analyses. Subgroup analyses were conducted (Table 4) comparing total stimulation time between studies, favouring shorter stimulation times (*z*(1) = 0.940, *p* = 0.347, Hedges’ *g* = 0.300, and 95% CI = −0.325–0.925).

## 7. Discussion

This study (Part I) aimed to determine the effects of PES and NMES in people with OD without excluding populations based on medical diagnoses. To base findings on the highest level of evidence, only RCTs were included. This systematic review and meta-analysis were conducted using PRISMA procedures as a guide. 

### 7.1. Systematic Review Findings

When comparing RCTs in pharyngeal and neuromuscular electrical stimulation (i.e., PES and NMES), various methodological problems became apparent. Some studies did not define OD or used divergent definitions, whereas other studies applied different inclusion criteria. Most studies included patients with confirmed OD by instrumental assessment, but several studies used screening, patient self-report or clinical assessments instead. Consequently, participant characteristics may differ widely between studies. Despite most studies included stroke patients, meta-analysis comparing diagnostic groups other than stroke was possible for NMES, however this could not be conducted for PES.

Furthermore, the great variety in outcome measures also restricted comparisons by meta-analysis. As heterogeneity between studies indicates that no estimated overall effect by meta-analysis should be determined, combining studies targeting different domains within the area of OD will have similar implications. For instance, meta-analyses based on both patients’ self-reported health-related quality of life and visuoperceptual evaluation of instrumental assessments would very likely lead to inappropriate estimated overall effects. Thus, to reduce heterogeneity between outcome measures, some studies were excluded from the meta-analysis. This strong focus on reducing heterogeneity between studies when performing meta-analysis also implies that data other than the authors’ primary outcomes may have been preferably included in this analysis. For example, the primary outcome for Dziewas, Stellato, Van Der Tweel, Walther, Werner, Braun, Citerio, Jandl, Friedrichs, Nötzel, Vosko, Mistry, Hamdy, McGowan, Warnecke, Zwittag and Bath [59] and Suntrup, Marian, Schröder, Suttrup, Muhle, Oelenberg, Hamacher, Minnerup, Warnecke and Dziewas [64] was readiness for decannulation, which was considered too different from outcomes in the other included studies. 

All eight PES studies compared neurostimulation with sham stimulation. However, among the 30 NMES studies, the comparison group variably consisted of usual care, DT, another dysphagia treatment or a combination of treatments. In contrast to PES studies that did not include any DT groups, most NMES studies combined neurostimulation with simultaneous DT. However, DT consisted of a wide range of behavioural interventions, using different treatment dosages, timings, and durations. Moreover, DT was referred to by many different names and acronyms (e.g., dysphagia training, behavioural intervention, classic treatment, or standard care). This suggest that care should be taken with the use of DT as an overarching term to group many different behavioural interventions to estimate overall effect sizes in meta-analyses. 

Furthermore, RCTs are characterised by random allocation of participants to intervention groups and blinding or masking the nature of treatment for participants. However, in neurostimulation studies, blinding is frequently not feasible and participants may identify what treatment arm they have been assigned to (e.g., the presence of neurostimulation equipment, the experience of active stimulation). Also, since neurostimulation thresholding in PES is frequently applied in all groups to mask treatment assignment, patients receiving sham stimulation would still have been exposed to a certain level of neurostimulation during thresholding. Those studies not using thresholding in sham groups (e.g., [59,64]) might show larger treatment effect differences when comparing neurostimulation versus sham stimulation.

### 7.2. NMES

When considering meta-analyses for NMES, the highest effect sizes were found for stroke populations. As existing reviews in NMES [10,12,18,19] excluded other patient populations, no comparisons could be made between clinical populations. In addition, only two reviews conducted meta-analyses [18,19] selecting studies using different inclusion criteria (e.g., excluding comparison groups with active treatment components [18] or excluding chronic stroke patients [19]). Reviews may also prefer different outcome data for meta-analyses, especially in the case of RCTs using a large battery of assessments. As such, total numbers of included studies vary per review, but comparisons between reviews may be falsely estimated due to differences in methodology.

In this systematic review, a wide range in effect sizes was found in NMES RCTs depending on outcome measures used. However, oral intake scales showed highest effects sizes when compared to visuoperceptual evaluation of instrumental assessment or clinical assessment. This might be explained by NMES treatment usually taking place over consecutive weeks, in contrast to other neurostimulation techniques (e.g., PES or rTMS) that may be restricted to limited sessions over a few days only. 

The great heterogeneity between DT groups also impeded comparisons between NMES only, NMES plus DT, and DT-only groups. No RCTs provided adequate DT group data to be included in the meta-analysis. For NMES groups, only two studies were included. As a result, information about the effects of DT is lacking. The negligible effect sizes found for NMES without DT were based on only two studies and the moderate effect sizes for combined NMES and DT were based on a total of seven studies.

Most studies performed NMES at motor stimulation level, whereas only a few studies included a group receiving NMES at sensory stimulation level. As none of these latter studies could be included in meta-analyses, no further details are available on comparisons between effect sizes for sensory versus motor stimulation. Also, terminology was confusing as sensory stimulation was sometimes referred to as sham stimulation [39].

NMES studies showed marked variation in the technical parameters and protocols applied. When comparing electrode configurations, both hyoid and combined hyoid and suprahyoid configurations showed negligible effects, whereas infrahyoid configurations resulted in moderate effects. A study using patient-dependent configurations showed promising results as well [55]. However, it remained unclear which criteria were used to decide on individual configurations. Furthermore, reporting on many technical parameters proved to be either incomplete or unclear for several studies (e.g., data on pulse duration, pulse rate, or stimulation time). As technical parameters may depend on medical device manufacturers, comparisons between brands may be warranted. For example, when considering pulse duration, a clear distinction in effect sizes is found between one study using a lower pulse rate—indicating a negative effect size—versus eight studies using higher pulse rates with moderate effect sizes. 

### 7.3. PES

Compared to NMES, fewer PES studies were identified and thus a more limited meta-analysis was conducted. RCTs included stroke populations, except for one study that included patients with multiple sclerosis [63]. All studies compared active PES with sham treatment in stroke patients and used mostly visuoperceptual evaluation of radiographic recordings of the swallowing act as an outcome measure. Meta-analysis identified a non-significant post-intervention between-group total effect size in favour of PES. This finding seemed in line with findings by Chiang, Lin, Hsiao, Yeh, Liang and Wang [19], but this comparison is limited as it is based on only two studies. Additionally, Bath, Lee and Everton [18] reported that PES studies did not show an effect for many outcome measures (e.g., post-treatment proportions of participants with dysphagia, swallowing ability, penetration and aspiration scores or nutrition). However, in contrast to previous reviews, Cheng, Sasegbon and Hamdy [7] found a significant, moderate effect size in favour of PES when conducting meta-analysis. Again, inclusion criteria between reviews differed. For example, two studies [59,64] were excluded from meta-analysis in this review as well as the reviews by Chiang, Lin, Hsiao, Yeh, Liang and Wang [19] and Bath, Lee and Everton [18], but were included in the review by Cheng, Sasegbon and Hamdy [7]. This may have impacted the overall effect size as both PES studies showed significant treatment effects. 

### 7.4. Moderators

Differences between NMES and PES studies made comparisons between RCTs difficult and hindered meta-analyses. Studies used different participant inclusion criteria in relation to underlying medical diagnoses or chronicity of stroke and used a large variety of outcome measures covering different domains within the area of OD. Outcome measures may also lack responsiveness, thus lack sensitivity to change during treatment. Moreover, studies varied significantly in technical parameters of neurostimulation. The number of studies and participants restricted the ability of statistical analyses to consider how each variable may have impacted the effects of neurostimulation. 

Studies frequently neglected to report on potential moderators of stimulation effects in sufficient detail. For example, stroke severity and OD severity are inextricably linked and may moderate stimulation effects, yet only very few studies provided data on stroke severity. Similar problems occur when the chronicity of a stroke is not reported or the possibility of spontaneous recovery is ignored. This is especially true during NMES treatment, which may span a period of several weeks. In addition, no consensus was reached regarding the optimal moment for outcome measurement. Consequently, in this review, between-subgroup meta-analyses were conducted using post-intervention data only, so that the possibility of spontaneous recovery during the intervention period was taken into consideration. 

### 7.5. Limitations

Despite a rigorous reviewing process following PRISMA guidelines and the use of RoB 2 to reduce bias, this review is subject to some limitations. Only RCTs published in English were included in this current study. Thus, some RCTs may have been excluded based on language criteria when their findings could have contributed to the current meta-analysis. Furthermore, meta-analyses included mostly stroke studies, thereby not providing effect sizes for other diagnostic patient populations. However, the main limitation of this review originates from the high degree of heterogeneity between studies, making comparisons across studies challenging. As such, generalisations and meta-analyses should be interpreted with care.

## 8. Conclusions

Meta-analyses for RCTS in NMES found a significant, large pre-post intervention effect size and significant, small post-intervention between-group effect size in favour of NMES. For PES studies, the meta-analyses showed a significant, moderate effect size for pre-post intervention, whereas overall between-group analysis did not result in significant treatment effects. Based on these results, NMES seems to have a more promising outcome compared to PES. However, only careful generalisations and interpretations of these meta-analyses can be made due to the NMES studies showing high heterogeneity in protocols and experimental variables, including potential moderators, and featuring inconsistent methodological reporting.

There is a need for more RCTs with larger sample sizes in addition to the standardisation of protocols and guidelines for reporting. These changes would better facilitate comparisons of studies and help to determine intervention effects more definitively. Delphi studies involving international experts might allow for a consensus to be reached, thus supporting future research, comparability and generalisability.

## Figures and Tables

**Figure 1 jcm-11-00776-f001:**
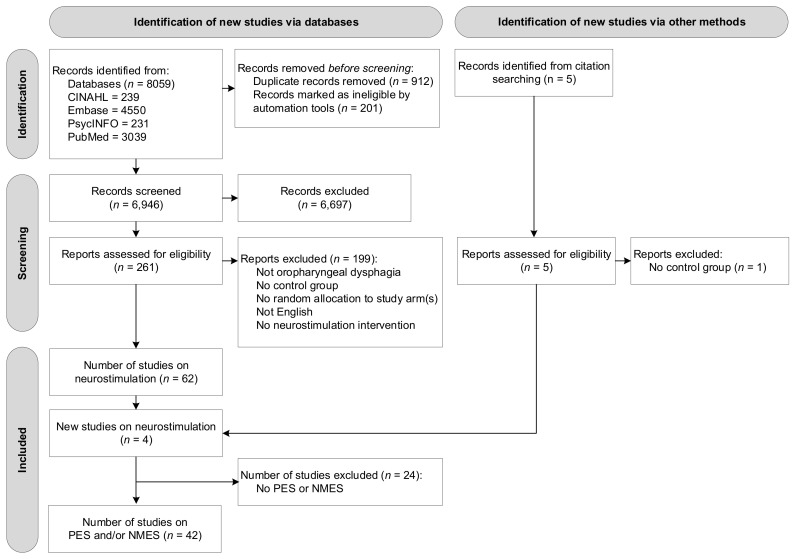
Flow diagram of the reviewing process according to the Preferred Reporting Items for Systematic Reviews and Meta-Analyses (PRISMA).

**Figure 2 jcm-11-00776-f002:**
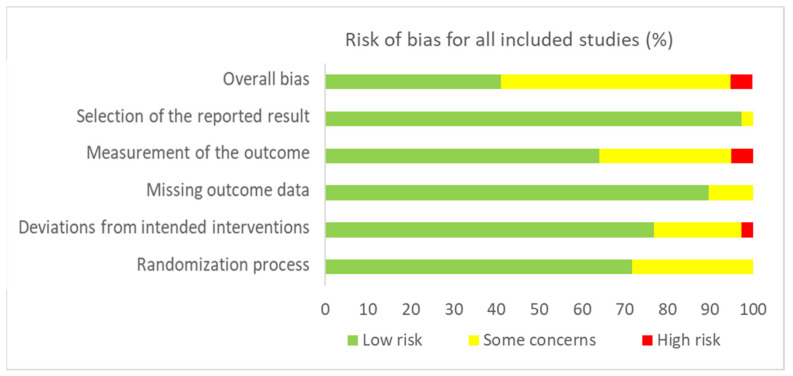
Risk of bias summary for all included studies (*n* = 42) in accordance with RoB-2.

**Figure 3 jcm-11-00776-f003:**
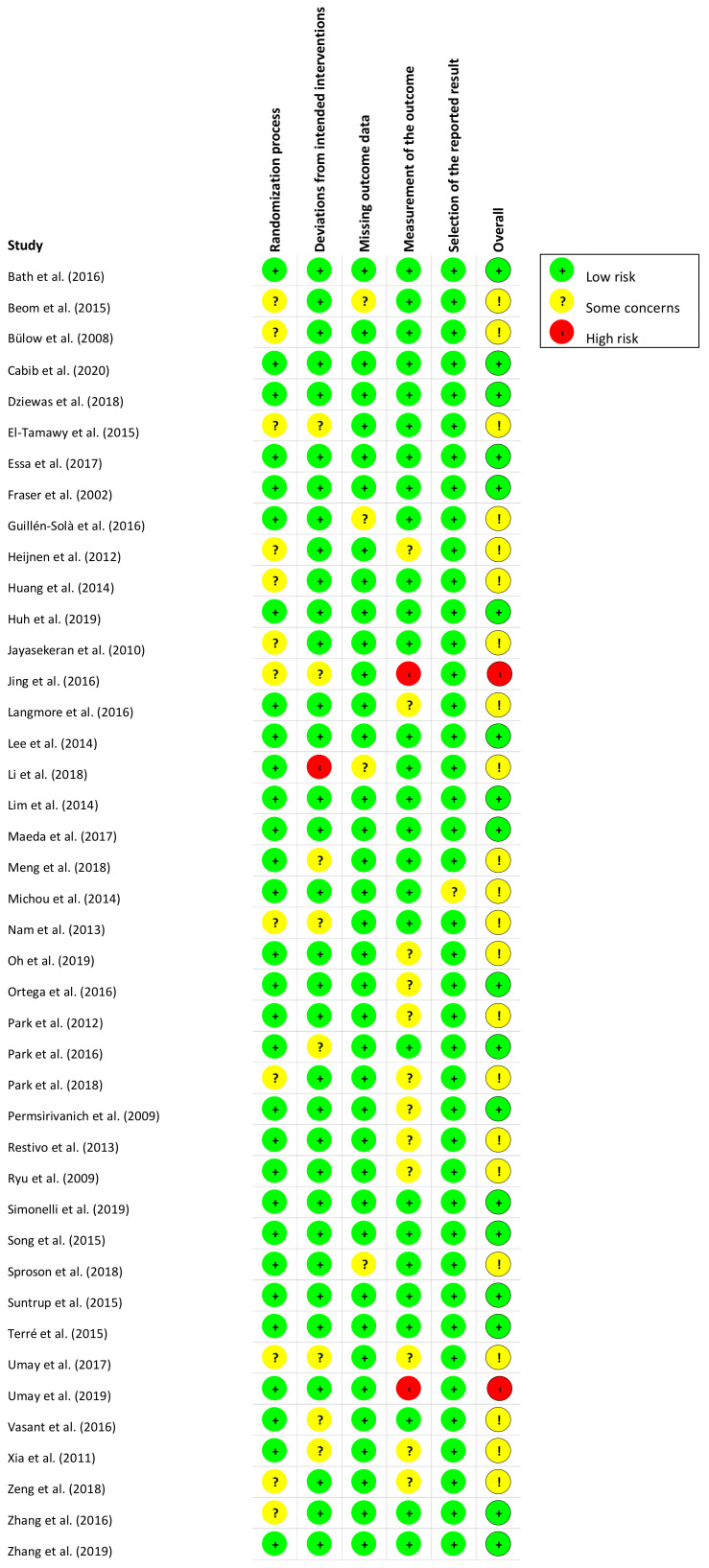
Risk of bias summary for individual studies (*n* = 42) in accordance with RoB-2.

**Figure 4 jcm-11-00776-f004:**
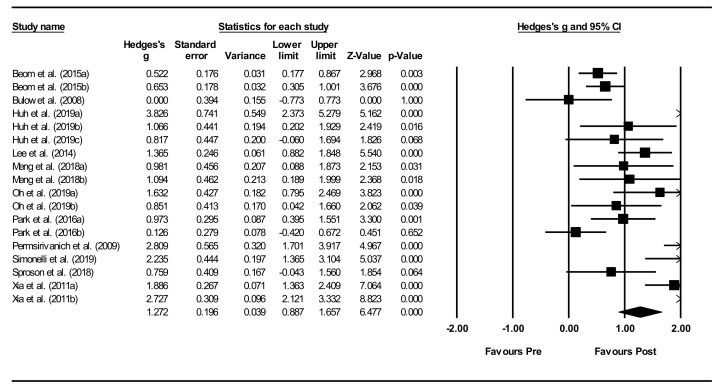
Neuromuscular electrical stimulation (NMES) within intervention group pre-post meta-analysis [28,29,34,37,40,42,45,47,49,51,55]. Note. Refer to Table 2 for explanation of the subgroups.

**Figure 5 jcm-11-00776-f005:**
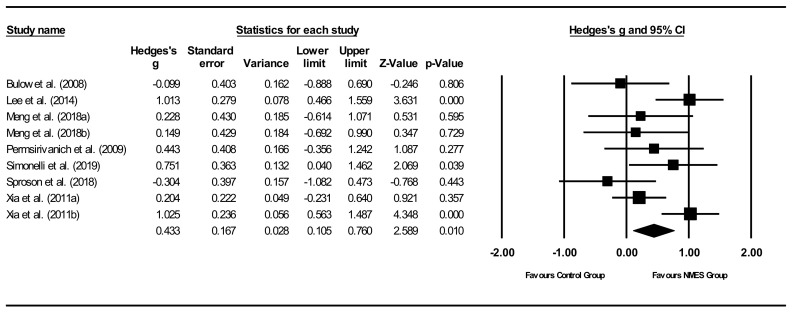
NMES between group post meta-analysis [29,37,40,47,49,51,55]. Note. Refer to Table 2 for explanation of the subgroups.

**Figure 6 jcm-11-00776-f006:**
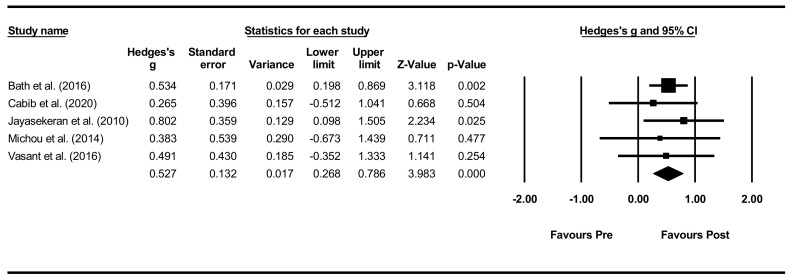
PES within intervention group pre-post meta-analysis [58,62,65,66,68].

**Figure 7 jcm-11-00776-f007:**
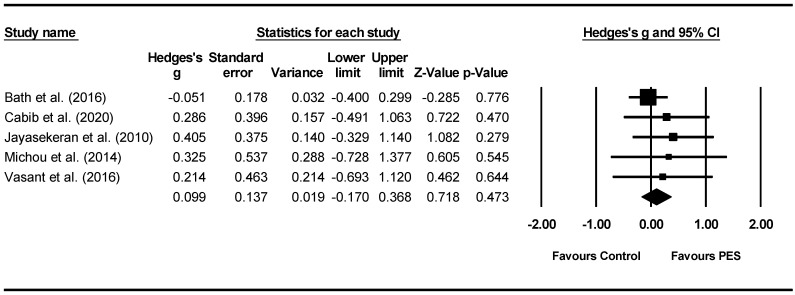
PES between group post meta-analysis [58,62,65,66,68].

**Table 1 jcm-11-00776-t001:** Search strategies.

Database and Search Terms	Number of Records
Cinahl: ((MH “Deglutition”) OR (MH “Deglutition Disorders”)) AND (MH “Randomized Controlled Trials”)	239
Embase: (swallowing/OR dysphagia/) AND (randomization/or randomized controlled trial/OR “randomized controlled trial (topic)”/OR controlled clinical trial/)	4550
PsycINFO: (swallowing/OR dysphagia/) AND (RCT OR (Randomised AND Controlled AND Trial) OR (Randomized AND Clinical AND Trial) OR (Randomised AND Clinical AND Trial) OR (Controlled AND Clinical AND Trial)).af.	231
PubMed: (“Deglutition”[Mesh] OR “Deglutition Disorders”[Mesh]) AND (“Randomized Controlled Trial” [Publication Type] OR “Randomized Controlled Trials as Topic”[Mesh] OR “Controlled Clinical Trial” [Publication Type] OR “Pragmatic Clinical Trials as Topic”[Mesh])	3039

**Table 2 jcm-11-00776-t002:** Study characteristics of studies on NMES and PES interventions for people with oropharyngeal dysphagia.

Study Country	Inclusion/Exclusion Criteria	Sample (N) Group	Group Descriptives (Mean ± SD) Age, Gender, Medical Diagnoses
**NeuroMuscular Electrical Stimulation (NMES) ^a^—*n* = *30***			
Beom, et al. [28] Country: Korea	OD as per VFSS Inclusion: stroke, traumatic brain injury or brain tumour >1 week ago; hemiplegia caused by hemispheric lesion; able to respond to pain Exclusion: no potential for recovery; severe communication difficulties; contraindications for neuromuscular electrical stimulation (NMES)	*n* = 132 Treatment group 1 (66), 50% NMES (Suprahyoid muscle stimulation) + DT [Denoted as ‘Beom et al. (2015a)’ in Figure 4] Treatment group 2 (66), 50% NMES (suprahyoid and infrahyoid muscles stimulation) + DT [Denoted as ‘Beom et al. (2015b)’ in Figure 4]	Treatment group 1: Age 64.4 ± 12.0 50% male Location of lesion: cortex (29), subcortex (20), brainstem (16), cerebellum (1) Treatment group 2: Age 59.8 ± 15.9 66.6% male Location of lesion: cortex (29), subcortex (14), brainstem (19), cerebellum (4) NS difference between groups
Bülow, et al. [29] Country: Sweden, The Netherlands, France	OD as per VFSS Inclusion: 50–80 years old; hemispheric stroke > 3 months; ability to swallow: ability to communicate Exclusion: brainstem involvement; progressive cerebrovascular disease; other neurologic diseases such as ALS, MS, or Parkinson’s disease; patients with tumors of the swallowing apparatus + radiotherapy/surgery to the neck; patients with no pharyngeal swallow; nasogastric tube insitu	*n* = 25 Treatment group 1 (13), 52% DT Treatment group 2 (12), 48% NMES (suprahyoid and infrahyoid muscles stimulation) + Diet modification	Combined treatment groups data: 64% male Treatment group 1: Age 71 (SD not reported) Treatment group 2: Age 70 Statistical difference between groups = NR
El-Tamawy, et al. [30] Country: Egypt	OD as per bedside screening (confirmed by VFSS once enrolled) Inclusion : acute stroke, severe dysphagia; able to ambulate; normal attention and communication skills; no other neurological disease; able to perform sit to stand test Exclusion : disturbed level of consciousness; dementia; psychiatric disorders; syncope; previous operation or injury to the head and neck area	*n* = 30 Treatment group 1 (15), 50% NMES + DT + (Medical treatment) Control group 2 (15), 50% Medical treatment	Treatment group 1: Age 61.5 ± 7.3 Control group 2: Age 61.3 ± 6.6 No further details on subjects within the groups.
Guillén-Solà, et al. [31] Country: Spain	OD as per VFSS (PAS ≥ 3) Subacute ischaemic stroke (1–3 weeks) Exclusion : Cognitive impairment (Short portable Mental Status Qnr >3), previous neurological diseases with risk of dysphagia	*n* = 62 Treatment group 1 (21), 33.9% DT Treatment group 2 (20), 32.2% DT + inspiratory and expiratory muscle training Treatment group 3 (21), 33.9% NMES + DT + sham inspiratory and expiratory muscle training	Treatment 1: Age = 68.9 ± 7 Male = 57.1% Treatment 2: Age = 67.9 ± 10.6 Male = 76.2% Treatment 3: Age = 70.3 ± 8.4 Male = 47.6% NS differences between groups
Heijnen, et al. [32] Country: The Netherlands	OD as per clinical assessment by SLT/VFSS Inclusion: 40–80 year olds with idiopathic Parkinson’s disease; ‘stable’ condition; unaltered antiparkinsonian medication protocol for ≥2 months Exclusion: other neurological disease; severe mental depression or cognitive degeneration (MMSE < 23); deep brain stimulation, malignancies, extensive surgery, radiotherapy to the head&neck region; severe cardiopulmonary disease, epilepsy, carotid sinus syndrome, dermatological diseases in head&neck area; dysphagia treatment in the preceding 6 months	*n* = 85 Treatment group 1 (28), 32.9% DT Treatment group 2 (27), 31.8% NMES at motor level + DT Treatment group 3 (30), 35.3% NMES at sensory level + DT	Treatment group 1: median age 69 78.6% male Treatment group 2: median age 65 74.1% male Treatment group 3: median age 66 76.7% male NS differences between groups
Huang, et al. [33] Country: Taiwan	OD as per 100 mL water test + SLT assessment Inclusion: recent cerebral hemispheric stroke; FOIS ≤ 4 Exclusion: impaired communication ability; dysphagia caused by other disease; use of an electrically sensitive biomedical device (eg. cardiac pacemaker); pneumonia or acute medical condition	*n* = 29 Treatment group 1 (11), 37.9% DT Treatment group 2 (8), 27.6% NMES Treatment group 3 (10), 34.5% NMES + DT	Treatment group 1: Age 67.0 ± 10.1 54.5% male Infarction (9); haemorrhage (2) Treatment group 2: Age 64.5 ± 14.4 62.5% male Infarction (6); haemorrhage (2) Treatment group 3: Age 68.9 ± 9.8 90% male Infarction (9); haemorrhage (1) NS differences between groups
Huh, et al. [34] Country: South Korea	OD as per VFSS Inclusion: stroke; sufficient cognitive and language skills to perform effortful swallow Exclusion: other neurological disease; contraindications to electrical stimulation	*n* = 31 Treatment group 1 (10), 32.3% NMES with horizontal electrodes configuration (supra and infrahyoid muscles stimulation) + DT (effortful swallow) [Denoted as ‘Huh et al. (2019a)’ in Figure 4] Treatment group 2 (11), 35.5% NMES with horizontal + vertical electrodes configuration (supra and infrahyoid muscles stimulation) + DT (effortful swallow) [Denoted as ‘Huh et al. (2019b)’ in Figure 4] Treatment group 3 (10), 32.3% NMES with vertical electrodes configuration (supra and infrahyoid mus cles stimulation) + DT (effortful swallow) [Denoted as ‘Huh et al. (2019c)’ in Figure 4]	Treatment group 1: Age 64.8 ± 14.1 90% male Infarction 6, haemorrhage 4 Treatment group 2: Age 60.45 ± 16.2 72.7% male Infarction 4, haemorrhage 7 Treatment group 3: Age 62.40 ± 12.7 50% male Infarction 4, haemorrhage 6 NS differences between groups
Jing, et al. [35] Country: China	OD as per Rattans dysphagia classification criteria, conducted by a rehabilitation nurse Inclusion: stroke; dysphagia (grade ≤ 5) within 1–3 days post stroke; no previous rehabilitation training; stable vital signs; signed informed consent Exclusion: not available	*n* = 60 Treatment group 1 (30), 50% NMES + DT (+ Medical treatment) Treatment group 2 (30), 50% DT (+ Medical treatment)	Treatment group 1: Age 67.9 ± 11.4 63.3% male 63% unilateral, 47% bilateral stroke, 70% infarction, 30% haemorrhage Treatment group 2: Age 68.6 ± 12.5 53.3% male 70% unilateral, 30% bilateral stroke, 77% infarction, 23% haemorrhage Statistical difference between groups = NR
Langmore, et al. [36] Country: USA	OD as per VFSS Inclusion: >21 year old patients ≥ 3 months post a full dose (≥50 Gy) of (chemo)radiotherapy for head&neck cancer, cancer free, severe dysphagia (PAS ≥ 4 on VFSS) Exclusion: dysphagia due to other cause, prior use of electrical stimulation, neurologic disease, presence of pacemaker/defibrillator, floor of mouth resection, inability to follow the study protocol	*n* = 127 Treatment group 1 (91), 71.7% NMES + DT Sham/treatment group 2 (36), 28.3% Sham NMES + DT	Treatment group 1: Age 62.1 + 9.2 86.2% male RT site: Oral—9.5%, Nasopharynx—8.6%, Oropharynx—47.4%, Hypopharynx—12.1%, Larynx—11.2%, Other—12.1% Stage: 1—7.4% 2—7.4% 3—21.1% 4—64.2% Sham/Treatment group 2: Age 61.5 + 10.6 84.6% male RT site: Oral—5.9%, Nasopharynx—13.7%, Oropharynx—45.1%, Hypopharynx—17.6%, Larynx—17.6%, Other—7.8% Stage: 1—0% 2—15.2% 3—13% 4—71.7% NS difference between groups
Lee, et al. [37] Country: Korea	OD as per VFSS Inclusion: 18–80 years. Supratentorial ischaemic stroke; FOIS of ≤5 as per VFSS within 10days of stroke; Korean Mini-Mental State Examination (K-MMSE) ≥21; stable underlying disease process Exclusion: pre-existing dysphagia; previous stroke; unstable cardiopulmonary status, serious psychological disorder or epilepsy; tumour or radiotherapy of the head&neck region; prior swallowing therapy; unstable medical conditions that may interfere with VFSS	*n* = 57 Treatment group 1 (31), 54.4% NMES + DT Treatment group 2 (26), 45.6% DT	Treatment group 1: Age 63.5 ± 11.4 years 71% male Lesion location: right (13), left (18) Cortical (5), subcortical (26) Treatment group 2: Age: 66.7 ± 9.5 years 77% male Lesion location: right (11), left (15) Cortical (4), subcortical (22) NS difference between groups
Li, et al. [38] Country: China	OD as per meeting the criteria for diagnosis of dysphagia post stroke. Recruitment through newspaper advertisements and flyers Inclusion: 50–80 year olds > 3 months post hemispheric stroke (first or recurrence); ability to elicit some swallow as per hyoid excursion or pharyngeal constriction on videographic swallow, ability to communicate, stable disease Exclusion: brainstem lesion or progressive neurological disease, presence of nasogastric tube, tumour, surgery of radiotherapy to the swallowing apparatus	*n* = 135 Treatment group 1 (45), 33.3% NMES + DT Treatment group 2 (45), 33.3% NMES Treatment group 3 (45), 33.3% DT 38, 40 and 40 patients in groups 1–3 respectively, completed the treatment. All descriptive data about groups given based on originally enrolled numbers.	Treatment group 1: Age 66.7 ± 14.6 53% male 44% haemorrhage, 56% infarct. Treatment group 2: Age 65.8 ± 13.2 49% male 49% haemorrhage, 51% infarct. Treatment group 3: Age 66.1 ± 13.1 51% male 47% haemorrhage, 53% infarct. NS differences between groups
Maeda, et al. [39] Country: Japan	OD as per VFSS Inclusion: ≥65 years; prescribed dysphagia rehabilitation for >3 weeks Exclusion: no cough provoked on exposure to a citric acid mist for <90 sec; inability to remain still for 15 min stimulation	*n* = 43 Treatment group 1 (22), 51.2% NMES at sensory level Sham group 2 (21), 48.8% Sham NMES	Treatment group: Age 82.7 ± 8.0 45.5% male Primary reason for admission: Dysphagia rehabilitation 63.6%; aspiration pneumonia 27.3%; other 9.1% Sham group 2: Age 86.0 ± 6.7 38.1% male Primary reason for admission: Dysphagia rehabilitation 42.9%; aspiration pneumonia 33.3%; other 23.8% NS differences between groups
Meng, et al. [40] Country: China	OD as per water swallow test (WST) by SLT, plus VFSS for those with the test score of grade II or above Inclusion: 18–85 year olds with stroke <6 months ago; alert, orientated, cooperative; dysphagia confirmed with VFSS Exclusion: presence of severe cardiac or pulmonary dysfunction, implanted cardiac pacemaker, dementia, aphasia; limited ability to follow instructions; severe aspiration; inability to swallow at all	*n* = 30 Treatment group 1 (10), 33.3% NMES (supra and infrahyoid muscles stimulation) + DT [Denoted as ‘Meng et al. (2018a)’ in Figure 4 and Figure 5] Treatment group 2 (10), 33.3% NMES (suprahyoid muscles stimulation) + DT [Denoted as ‘Meng et al. (2018b)’ in Figure 4 and 5] Treatment group 3 (10), 33.3% DT	Treatment group 1: Age 65.2 ± 10.7 70% male 80% infarction, 20% haemorrhage. Treatment group 2: Age 67.2 ± 15.8 60% male 70% infarction, 30% haemorrhage. Treatment group 3: Age 64.4 ± 9.0 70% male 70% infarction, 30% haemorrhage. NS differences between groups
Nam, et al. [41] Country: Korea	OD as per VFSS Inclusion: subacute stroke or brain injury; VFSS showing aspiration or penetration and decreased laryngeal elevation Exclusion: chronic dysphagia	*n* = 50 Treatment group 1 (25), 50% NMES (suprahyoid muscles stimulation) + DT Treatment group 2 (25), 50% NMES (supra and infrahyoid muscles stimulation) + DT	Treatment group 1: Age 62.3 ± 11.4 52% male Location of lesion: cortex 13, subcortex 6, brainstem 5, cerebellum 1 Treatment group 2: Age 60.9 ± 12.3 56% male Location of lesion: cortex 10, subcortex 8, brainstem 6, cerebellum 1 NS differences between groups
Oh, et al. [42] Country: Korea	OD as per VFSS Inclusion: post-stroke dysphagia for <6 months; presence of voluntary swallow; Korean MMSE score ≥20 Exclusion: implanted cardiac pacemaker; severe communication disorder; tracheostomy; Hx of seizure or epilepsy; unstable medical conditions; skin problems associated with electrode placement	*n* = 26 Treatment group 1 (14), 54% [Denoted as ‘Oh et al. (2019a)’ in Figure 4] NMES (suprahyoid muscles stimulation) + DT Treatment group 2 (12), 46% NMES (infrahyoid muscles stimulation) + DT [Denoted as ‘Oh et al. (2019b) in Figure 4]	Treatment group 1: Age 56.3 ± 13.3 50% male Site of stroke lesion: middle cerebral artery (8), midbrain (1), frontal lobe (2), internal capsule (2), corona radiate (1) Treatment group 2: Age 58.7 ± 14.8 41.7% male Site of stroke lesion: middle cerebral artery (7), midbrain (1), frontal lobe (1), internal capsule (2), corona radiate (1) NS differences between groups
Ortega, et al. [43] Country: Spain	OD as per VFSS Inclusion: ≥70 years with oropharyngeal dysphagia; PAS > 2 Exclusion: active neoplasm or infectious process; epilepsy or convulsive disorders; gastroesophageal reflux disease; implanted electrodes or pacemakers; severe dementia; current participation in another trial	*n* = 38 Treatment group 1 (19), 50% Chemical sensory stimulation with TRPV1 agonist (capsaicin) Treatment group 2 (19), 50% NMES	Treatment group 1: Age 81.2 ± 5.6 42.1% male Dysphagia cause: elderly (8), stroke (8), neurodegenerative disease (3) Treatment group 2: Age 79.8 ± 4.8 47.4% male Dysphagia cause: elderly (7), stroke (8), neurodegenerative disease (3) NS difference between groups.
Park, et al. [44] Country: Korea	OD as per VFSS Inclusion: >1 month post stroke, adequate cognition. Exclusion: SAH, carotid stenosis, unable to perform NMES (as per observation and palpation)	*n* = 18 Treatment group 1 (9), 50% NMES + DT (Effortful swallow) Treatment group 2 (9), 50% NMES at sensory level + DT (Effortful swallow)	Treatment group 1: Age 68.7 ± 12.8 56% intracranial haemorrhage (ICH), 44% infarct. Treatment group 2: Age 62.0 ± 17.2 78% ICH, 22% infarct. Gender reported at cohort level: 88% male. NS differences between groups
Park, et al. [45] Country: Korea	OD as per VFSS Inclusion: stroke; onset >6months; able to swallow against resistance applied by electrical stimulation; able to actively participate; MMSE ≥ 24 Exclusion: psychiatric disorders or dementia; cardiac pacemaker; severe communication disorder; epilepsy; unstable medical condition; skin problems affecting electrode placement	*n* = 50 Treatment group 1 (25), 50% NMES + DT (Effortful swallow) [Denoted as ‘Park et al. (2016a)’ in Figure 4] Treatment group 2 (25), 50% NMES at sensory level + DT (Effortful swallow) [Denoted as ‘Park et al. (2016b)’ in Figure 4]	Treatment group 1: Age 54 ± 11.93 48% male Infarct = 14, haemorrhage = 11 Treatment group 2: Age 55.8 ± 12.23 56% male Infarct = 12, haemorrhage = 13 NS differences between groups
Park, et al. [46] Country: Korea	OD as per VFSS Inclusion: Parkinson’s disease, adequate cognition (MMSE score > 20); age < 75 years; ability to swallow voluntarily; and Hoehn and Yahr scale < 3 points Exclusion: other neurological disease: deep brain stimulation treatment; neck pain or neck surgery; implanted electronic devices; severe communication problem; severe dyskinesia of the head and neck; history of seizure/epilepsy	*n* = 18 Treatment group 1 (9), 50% NMES + effortful swallow Treatment group 2 (9), 50% NMES at sensory level + DT (effortful swallow)	Treatment group: Age 63.44 ± 13.55 55% male Treatment group 2: 54.67 ± 13.82 33% male *NB*. Patients’ medical diagnosis: stroke (Table 1) [Error?] NS differences between groups
Permsirivanich, et al. [47] Country: Thailand	OD as per VFSS Inclusion: stroke; dysphagia with safe swallow as per VFSS Exclusion: not listed	*n* = 23 Treatment group 1 (11), 48% DT (full program) Treatment group 2 (12), 52% NMES + DT (restricted program)	Treatment group 1: Age 64.7 ± 9.4 36.4% male Type of stroke: infarction 81.8%, haemorrhage 18.2% Treatment group 2: Age 64.5 ± 8.8 41.7% male Type of stroke: infarction 75%, haemorrhage 25% NS differences between groups
Ryu, et al. [48] Country: Korea	OD confirmed via VFSS and patients on a restricted diet Inclusion: surgical with or without (chemo)radiation for head and neck cancer, stable vital signs, able to participate in treatment Exclusion: <20 years, cognitive impairment, history of cerebrovascular disease, serious psychologic disorder, cardiac pacemaker, unable to tolerate electrical stimulation	*n* = 26 Treatment group 1 (14), 53.8% NMES + DT Sham/Treatment group 2 (12), 46.2% Sham NMES + DT	Treatment group 1: Age 63.4 ± 7.3 100% male Larynx ca = 6 Hypopharynx ca = 3 Oropharynx = 4 Oral =1 T1-T2 = 6 T3-T4 = 8 Sham/Treatment group 2: Age 60.8 ± 12.0 92% male Larynx ca = 5 Hypopharynx ca = 1 Oropharynx = 4 Oral =2 T1-T2 = 7 T3-T4 = 4 Unknow*n* = 1 Statistical difference between groups = NR
Simonelli, et al. [49] Country: Italy	OD definition as per clinical swallow exam (confirmed by instrumental exam) Inclusion: Age: 18–85 years; first-time stroke (confirmed by MRI); presence of dysphagia for 3 weeks > 3 months, with preservation of cough reflex; feeding tube-dependence, FOIS ≤ 2; stable underlying disease process Exclusion: Cognitive impairment or mental depression; concomitant neurodegenerative disease; unstable cardiopulmonary status; head & neck tumour, surgery, or radiotherapy; cardiac pacemaker or history of seizures or epilepsy; previous swallowing therapy	*n* = 33 Treatment group 1 (17), 51.5% NMES + DT Treatment group 2 (16), 48.5% DT	Treatment group 1: Age 67.2 ± 16.2 62.5% male (10?) Left CVA (4), right CVA (6), Other (6) Treatment group 2: Age 72.4 ± 12.3 37.5% male (6) Left CVA (6), right CVA (6), Other (3) NS differences between groups.
Song, et al. [50] Country: Korea	OD as per VFSS or rehabilitation doctor Inclusion: Cerebral palsy (CP) diagnosis by rehabilitation doctor Exclusion: Vision or hearing disorders, seizure disorders, pacemaker	*n* = 20 Treatment group 1 (10), 50% NMES + DT (Oral sensorimotor treatment) Sham/Treatment group 2 (10), 50% Sham NMES + DT (Oral sensorimotor treatment)	Treatment group 1: Age = 6.20 ± 2.78 70% male CP type: Hemiplegia = 2 Diplegia = 5 Quadriplegia = 3 Flaccid = 0 Sham/Treatment group 2: Age = 6.00 ± 2.40 60% male CP type: Hemiplegia = 4 Diplegia = 3 Quadriplegia = 2 Flaccid = 1 NS differences between groups.
Sproson, et al. [51] Country: UK	OD as per VFSS Inclusion: medically stable; >1 month post-stroke; no other neurological disease; dysphagia incorporating reduced laryngeal elevation (confirmed by VFSS) Exclusion: <18 years; pacemaker; serious cardiac disease; severe cognitive/communication difficulties; lesions/infections in the treatment site	*n* = 24 Treatment group 1 (12), 50% NMES + DT Usual care group 2 (12), 50% Usual care (Different from DT)	Treatment group 1: Age 76 ± 11.4 67% male 33% >1 stroke event Time post-stroke 17.3 months ± 25.0 Usual care group 2: Age 79 ± 11.4 66.7% male 33% >1 stroke event Time post-stroke 9.1 months ± 20.5 Significant difference between groups = NR.
Terré and Mearin [52] Country: Spain	OD as per VFSS demonstrating aspiration Inclusion: >18 years, acquired brain injury (stroke, TBI); <6 months since insult; able to understand and follow instructions for treatment; medically stable Exclusion: previous stroke or TBI; previous dysphagia secondary to other aetiology; no other metabolic or neurological diseases	*n* = 20 Treatment group 1 (10), 50% NMES + DT Sham/Treatment group 2 (10), 50% Sham NMES + DT	Treatment group 1: Age 46.0 ± 16 60% male 70% stroke, (haemorrhagic = 5, ischaemic = 2) 30% TBI Sham/Treatment group 2: Age 51 ± 23 60% male 70% stroke (haemorrhagic = 6, ischaemic = 1) 30% TBI Significant difference between groups = NR.
Umay, et al. [53] Country: Turkey	OD as per clinical swallow evaluation and FEES Inclusion: aged 45–75 years, <1 month post stroke (MRI confirmed), admitted to rehabilitation hospital Exclusion: haemorrhagic infarction or bilateral involvement, malignancy, head and/or neck surgery, previous stroke, pulmonary or swallowing disorder, gastroesophageal reflux, dementia or psychiatric disorder, and smoking	*n* = 98 Treatment group 1 (58), 59% NMES at sensory level + DT Sham/Treatment group 2 (40), 41% Sham NMES + DT	Treatment group 1: Age 61.03 ± 10.05 70.7% male 87.9% middle cerebral artery (MCA) stroke, 12.1% posterior inferior cerebellar (PICA) stroke Sham/Treatment group 2: Age 62.40 ± 9.93 87.5% male 87.5% middle cerebral artery (MCA) stroke, 12.5% posterior inferior cerebellar (PICA) stroke NS difference between groups.
Umay, et al. [54] Country: Turkey	OD as per Paediatric Eating Assessment Tool-10 and FEES. Inclusion: Children aged 2–6 years with cerebral palsy Exclusion: maxillary, head or neck surgery or botulinum toxin treatment, structural oropharyngeal abnormality, oesophageal dysphagia and/or gastroesophageal reflux disease, medical and/or physical therapy for dysphagia, severe cognitive, visual, auditory, and sensory impairments, drug use due to seizure or spasticity, serious pulmonary or cardiac disease, bleeding risk	*n* = 102 Treatment group 1 (52), 51% NMES at sensory level + DT Sham/Treatment group 2 (50), 49% Sham NMES + DT	Treatment group 1: Age 51.97 ± 24.46 months 56% male Motor function status as per GMFCS (I = walks with no limitations, V = wheelchair). I = 0 II = 7 III = 10 IV = 22 V = 13 Sham/Treatment group 2: Age 47.95 ± 23.18 months 46% male I = 0 II = 11 III = 11 IV = 16 V = 12 NS difference between groups.
Xia, et al. [55] Country: China	OD as per Standardised Swallow Assessment (SSA) and VFSS Inclusion: cerebral infarction or haemorrhage (diagnosed by CT or MRI); no pulmonary diseases; 40–80 years old; cognitively intact and able to cooperate Exclusion: None	*n* = 120 Treatment group 1 (40), 33.3% DT Treatment group 2 (40), 33.3% NMES [Denoted as ‘Xia et al. (2011a)’ in Figure 4 and 5] Treatment group 3 (40), 33.3% NMES + DT [Denoted as ‘Xia et al. (2011b)’ in Figure 4 and Figure 5]	Treatment group 1: Age 65.32 ± 14.29 62.5% male 42.5% haemorrhage, 45% infarct, 12.5% other stroke. Treatment group 2: Age 66.40 ± 15.63 57.5% male 35% haemorrhage, 55% infarct, 10% other stroke. Treatment group 3: Age 65.85 ± 14.63 70% male 32.5% haemorrhage, 62.5% infarct, 0.5% other stroke. NS difference between groups.
Zeng, et al. [56] Country: China	OD as per Kubota water-drinking test Inclusion: first-onset stroke (confirmed via MRI); able to actively cooperate; no significant cognitive disorder, aphasia, or other diseases affecting understanding Exclusion: critical condition or vital organ failure; cardiac pacemaker; metal implants or internal orthotics; comorbidities of malignant tumours, skin damage, heart disease, acute seizure/epilepsy, peripheral nerve damage	*n* = 112 Treatment group 1 (59), 52.7% DT Treatment group 2 (53), 47.3% NMES + DT	Treatment group 1: Age 66.13 ± 13.03 73.5% male NIHSS score = 4.25 ± 2.45 Treatment group 2: Age 67.92 ± 12.31 69.4% male NIHSS score = 5.02 ± 2.32 NS differences between groups at baseline.
Zhang, et al. [57] Country: China	OD as per VFSS Inclusion: primary diagnosis of medullary infarction confirmed via CT/MRI); onset <1 month; age 40–80 years; no severe cognitive impairment Exclusion: unstable vital signs, inflammatory markers; cardiac pacemaker or other electrical implants; dysphagia caused by structural lesions; skin lesions or metal implants in area of treatment; a history of epilepsy, malignancies, or other neurologic disease; pregnancy; spastic paralysis	*n* = 82 Treatment group 1 (27), 32.9% DT Treatment group 2 (28), 34.2% NMES at sensory level + DT Treatment group 3 (27), 32.9% NMES at motor level + DT	Treatment group 1: Age 62.6 ± 8.7 62.9% male Time since infarct: 21.3 ± 4.1 days Treatment group 2: Age 61.3 ± 7.1 57.1% male Time since infarct: 22.1 ± 4.0 days Treatment group 3: Age 62.2 ± 9.2 70.3% male Time since infarct: 20.6 ± 4.3 days NS differences between groups.
**Pharyngeal Electrical Stimulation (PES)—*n* = *8***			
Bath, et al. [58] Country: UK, Spain, Germany, Denmark, France	OD as per Toronto bedside swallowing screening test (TorBSST) fail + VFSS with PAS ≥ 3 Inclusion: stroke (ischaemic or haemorrhagic); >18 years; alert/rousable Exclusion: previous dysphagia, dysphagia due to another condition, implanted pacemaker/compromised cardio-pulmonary status, receiving oxygen, advanced dementia, distorted oropharyngeal anatomy, pregnant/breastfeeding mother	*n* = 162 Treatment group: (87) 54% PES Sham group: (75) 56% Sham PES	Treatment group: Age = 74.4 ±11.2 Male = 55.2% Ischaemic stroke = 89.5% Haemorrhagic = 10.5% PAS >2 in 90.8% Sham group: Age = 74.9 ± 12.6 Male = 61.3% Ischaemic stroke = 88% Haemorrhagic = 10.7% (Non-stroke = 1.3%) PAS > 2 in 92% NS difference between groups
Dziewas, et al. [59] Country: Germany, The Netherlands, Italy, Austria, UK	OD not defined Inclusion: ≥18 years, tracheostomy due to severe dysphagia after stroke (haemorrhagic or ischaemic); minimum 48 h mechanical ventilation, sedation free (min 3 days), Richmond Agitation Sedation Scale (RASS > −1) Exclusion: infratentorial stroke, pre-existing dysphagia, or other diseases causing dysphagia; participation in other study affecting PES, presence of a cardiac pacemaker/implantable defibrillator, nasal deformity, previous oesophageal surgery, any difficult/unsafe nasogastric tube placement, need for high levels of oxygen supply (>2 L/min), emergency treatment, or <3 months’ life expectancy	*n* = 69 Treatment group (35), 50.7% PES Sham group (34), 49.3% Sham PES 2nd open label treatment: Delayed group (*n* = 30) - Sham group still with a tracheostomy received late treatment; Retreat group (*n* = 16) PES group still with a tracheostomy received a 2nd treatment	Treatment group: Age = 61.7 ± 13 Male = 69% Sham group: Age = 66.8 ± 10.3 Male = 59% NS differences between groups
Essa, et al. [60] Country: UK	OD as per VFSS or FEES. Inclusion: First stroke, anterior cerebral circulation or brainstem; ≤6 weeks post onset; medically stable Exclusion: advanced dementia; other neurological reasons for dysphagia; previous dysphagia; cardiac pacemaker or defibrillator; compromised cardiac or respiratory status; significant structural abnormalities of the mouth or throat	*n* = 16 Treatment group (8), 50% Pharyngeal electrical stimulation (PES) Sham group (8), 50% Sham PES	Treatment group: Age 58.6 ± 13.4 62.5% male Stroke type: infarct (7), bleed (1) Sham group: Age 70.5 ± 11.8 62.5% male Stroke type: infarct (7), bleed (1) NS differences between groups.
Fraser, et al. [61] Country: UK	OD as per VFSS Inclusion: acute hemispheric stroke Exclusion: no details given	*n* = 16 Treatment group (10), 62.5% PES Sham group (6), 37.5% Sham PES	Descriptive statistics only Treatment group: Age range 65–93 60% male Sham group: Age range 56–78 66.6% male Statistical difference between groups = NR
Jayasekeran, et al. [62] Country: UK	OD as per VFSS >3. Inclusion: healthy volunteers for protocol 1; study 2—admitted with anterior circulation cerebral infarct or haemorrhage Exclusion: dementia, pacemaker or implantable cardiac defibrillator, unstable cardiopulmonary status, severe receptive aphasia, distorted oropharyngeal anatomy, dysphagia from conditions other than stroke	Protocol 1 (active or sham PES on virtual lesion) *n* = 11 (+2 for reversal of swallowing behaviour) Patients their own controls. Protocol 2 (PES with varying dose) *n* = 22 Group 0 (6), 27.2% Group 3 (4), 18.2% Group 5 (4), 18.2% Group 9 (4), 18.2% Group 15 (4), 18.2% Protocol 3 (active or sham PES in acute stroke) *n* = 28 Treatment group (16), 57% Sham group (12), 43%	Protocol 1: Age range 24–47 yrs 45.5% male (no data on treatment and sham groups separately) Protocols 2 and 3: Age 74 ± 10 68% male (No consistent data on treatment and sham groups separately for both protocols) Difference between groups NS
Restivo, et al. [63] Country: Italy	OD as per VFSS Inclusion: Patients with stable multiple sclerosis (MS) with dysphagia for >2 months; no dysphagia intervention in the preceding 3 months; >18 years; Expanded Disability Status Scale (EDSS) < 7.5 Exclusion: neurologic disease other than MS; age >60 years; concomitant illness or upper gastrointestinal disease; inability to give informed consent because of cognitive impairment	*n* = 20 Treatment group (10), 50% Pharyngeal stimulation Sham group (10), 50% Sham pharyngeal stimulation	Cohort demographics supplied, no group descriptives given. Mean age = 39.7 ± 6.5 years 35% male Relapsing-remitting MS = 14, Secondary progressive MS = 6 Mean EDSS = 5.7 ± 0.8; mean disease duratio*n* = 9.8 ± 2.4 years; mean dysphagia duratio*n* = 22.0 ± 7.4 months Statistical difference between groups = NR
Suntrup, et al. [64] Country: Germany	OD as per FEES Inclusion: tracheostomised, weaned off mechanical ventilation, unable to be decannulated due to severe persistent dysphagia Exclusion: pre-existing dysphagia; presence of implanted electronic devices of any kind	*n* = 30 Treatment group (20), 66.6% Pharyngeal stimulation Sham group (10), 33.3% Sham pharyngeal stimulation	Treatment group: Age 63.0 ± 14.5 years 45% male 90% ischaemic, 10% haemorrhagic stroke. 70% supratentorial, 30% infratentorial Sham group: Age: 66.7 ± 14.5 years 60% male 80% ischaemic, 20% haemorrhagic stroke. 90% supratentorial, 10% infratentorial Difference between groups NS
Vasant, et al. [65] Country: UK	OD as per TOR-BSST confirmed by MBS or FEES with most (but not all) patients Inclusion: dysphagia following anterior or posterior cerebral circulation infarct (ischemic and haemorrhagic) <6 weeks ago; medically stable at inclusion; no history of intubation/tracheotomy Exclusion: advanced dementia, other neurological conditions causing dysphagia, previous history of dysphagia, presence of cardiac pacemaker or implanted cardiac defibrillator, other severe cardiac or respiratory conditions, significant oral/pharyngeal structural abnormalities, continuous oxygen requirements	*n* = 35 at 2 weeks post treatment, *n* = 33 at 3 months post treatment. Treatment group (15), 48.4% PES Sham group (16), 51.6% Sham PES	Treatment group: (median) age = 71 Interquartile range (IQR) =56–79. 50% male NIHSS: median score = 10.0 (IQR= 5.2, 18.5) Sham group: (median) age = 71 (IQR = 61–78) 72% male NIHSS: median score = 12.5 (IQR = 9.2, 16.8) No other stroke, site of lesion details reported. NS differences between groups.
**Combined Neurostimulation Interventions—*n* = *4***			
Cabib, et al. [66] Country: Spain	OD as per VFSS Inclusion: >3 months post unilateral stroke, stable medical condition Exclusion: neurodegenerative disorders, epilepsy, drug dependency, brain or head trauma or surgery, structural causes of OD, pacemaker or metallic body implants, and pregnancy or lactation	*n* = 36 Treatment group 1 (12), 33.3%. rTMS Treatment group 2 (12), 33.3% Capsaicin Treatment group 3 (12), 33.3% PES	Treatment group 1: Age 70.0 ± 8.6 75% male 0% haemorrhage, 100% infarct. Treatment group 2: Age 74.3 ± 7.8 58% male 8% haemorrhage, 92% infarction Treatment group 3: Age 70.0 ± 14.2 92% male 25% haemorrhage, 75% infarction Difference between groups NS
Lim, et al. [67] Country: Korea	OD as per VFSS Inclusion: primary diagnosis unilateral cerebral infarction or haemorrhage (CT or MRI); stroke onset <3 months; patients who could maintain balance during evaluation + treatment; and adequate cognitive function to participate Exclusion: could not complete VFSS/failed the examination; presence of dysphagia pre stroke; history of prior stroke, epilepsy, tumor, radiotherapy in the head and neck, or other neurological diseases;unstable medical condition; and contraindication to magnetic or electrical stimulation	*n* = 47 Treatment group 1 (15), 32% DT Treatment group 2 (14), 30% DT + rTMS Treatment group 3 (18), 38% DT + NMES	Treatment group 1: Age 62.5 ± 8.2 60% male 34% haemorrhage, 66% infarction Treatment group 2: Age 59.8 ± 11.8 43% male 71% haemorrhage, 29% infarction Treatment group 3: Age 66.3 ± 15.4 67% male 66% haemorrhage, 44% infarction Difference between groups NS
Michou, et al. [68] Country: UK	OD as per diagnoses made by SLT (confirmed with VFSS at start of treatment) Inclusion: post stroke dysphagia for >6 weeks Exclusion: Hx of dementia, cognitive impairment, epilepsy, head&neck surgery; neurological defects prior to stroke; cardiac pacemaker or defibrillator in-situ; severe concomitant medical conditions; structural oropharyngeal pathology; intracranial metal; pregnancy; medications acting on CNS	*n* = 18 Treatment group 1 (6), 33.3% Pharyngeal electrical stimulation (PES) Treatment group 2 (6), 33.3% Paired associative stimulation (PAS) Treatment group 3 (6), 33.3% Repetitive transcranial magnetic stimulation (rTMS)	Treatment group: Avg age 60.3 83% male Treatment group 2: Avg age 67.3 100% male Treatment group 3: Avg age 67.8 66.7% male Overall: 63 ± 15 weeks post stroke with 7.6 ± 1 on NIHHS Statistical difference between groups = NR
Zhang, et al. [69] Country: China	OD as per DOSS by a well trained doctor Inclusion: stroke as per MRI <2 months earlier; aged 50–75 yrs; normal consciousness, stable vital signs, presence of dysdipsia and dysphagia Exclusion: brain trauma or other central nervous system disease; unstable arrhythmia, fever, infection, epilepsy, or use of sedative drugs; poor cooperation due to serious aphasia or cognitive disorders; contraindications to magnetic or electrical stimulation	*n* = 64 Treatment group 1 (16), 25%. Sham rTMS + NMES Treatment group 2 (16), 25% Ipsilateral rTMS + NMES Treatment group 3 (16), 25% Contralateral rTMS + NMES Treatment group 4 (16), 25% Bilateral rTMS + NMES	Treatment group 1: Age 55.9 ± 8.9 43% male 61.5% subcortical, 38.5% brainstem Treatment group 2: Age 56.8 ± 9.7 54% male 30.8% subcortical, 69.2% brainstem Treatment group 3: Age 56.5 ± 10.1 50% male 58.3% subcortical, 41.7% brainstem Treatment group 4: Age 53.1 ± 10.6 31% male 61.5% subcortical, 38.5% brainstem * All data given on participants that finished the trial and follow-up period (*n* = 52)

^a^ NMES is at motor stimulation level unless explicitly mentioned. Notes. CNS—central nervous system; CP—cerebral palsy; CT–computed tomography; CVA–cerebrovascular accident; DOSS–dysphagia outcome and severity scale; DT–dysphagia therapy; FEES–fiberoptic endoscopic evaluation of swallowing; FOIS–functional oral intake scale; ICH–intracranial haemorrhage; MMSE–Mini-Mental State Exam; MRI–magnetic resonance imaging; MS–multiple sclerosis; NIHSS–National Institutes of Health Stroke Scale; NMES–neuromuscular electrical stimulation; OD–oropharyngeal dysphagia; OST–oral sensorimotor treatment; PAS–penetration–aspiration score; PES–pharyngeal electrical stimulation; rTMS–repetitive transcranial magnetic stimulation; SAH–subarachnoid haemorrhage; sEMG–surface electromyography; SLT–Speech and Language Therapist; TBI–traumatic brain injury; tDCS–transcranial direct current stimulation; TOR-BSST–Toronto Bedside Swallowing Screening test; VFSS–videofluoroscopic swallowing study.

**Table 3 jcm-11-00776-t003:** Outcome of NMES and PES interventions for people with oropharyngeal dysphagia.

Study	Intervention Goal	Procedure, Delivery and Dosage Per Intervention Group	Outcome Measures	Intervention Outcomes/Conclusions
**NeuroMuscular Electrical Stimulation (NMES) ^a^—*n* = *30***				
Beom, et al. [28]	To investigate the effectiveness of NMES to suprahyoid muscle compared with NMES to infrahyoid muscle in brain-injured (stroke) patients with dysphagia	Procedure: NMES as per VitalStim therapy training manual 10–15 sessions, 30 min each, over 2–3 weeks DT during NMES sessions, as per videofluoroscopy swallow study (VFSS) Treatment group 1: NMES to the suprahyoid muscles (4 electrodes) 60 Hz pulse frequency, 500 ms pulse interval, using Stimplus Treatment group 2: NMES to the suprahyoid muscles (2 electrodes) and infrahyoid muscles (2 electrodes) 80 Hz pulse frequency, 700 ms pulse interval, using Vitalstim	Primary outcomes: FDS ^b^; SFS; aspiration/penetration based off VFSS pre and post treatment. Secondary outcome: N/R	No statistically significant differences between groups Both treatments showed significant improvement in FDS (*p* < 0.001) and SFS (*p* < 0.001), and non-significant improvements in penetration or aspiration
Bülow, et al. [29]	To evaluate and compare the outcome of NMES versus traditional swallowing therapy (TT) in stroke patients	Procedure: NMES as per VitalStim therapy training manual (Placement 3B) 15 sessions, 60 min each, 5 days/week for 3 weeks Diet modifications as per SLT recommendations Treatment group 1: NMES to supra & infra hyoid 4.5–25 mA (mean = 13 mA) Treatment group 2: clinician determined manoeuvers/ treatment techniques	Primary outcomes: Patient reported VAS (swallowing complaints); VFSS measure ^b^ (performed day of last treatment). Secondary outcome: N/R	No statistically significant differences between groups VAS = No significant improvement for NMES. Significant improvement (*p* < 0.01) noted for combined group effect VFSS parameters = No significant improvement for NMES nor combined group effect
El-Tamawy, et al. [30]	Assess the effect of NMES and physical therapy program on severe poststroke dysphagia	Treatment group 1: Standard medical treatment NMES: 30 min of 80 Hz frequency 0–150 V amplitude stimulation, intensity 0–25 mA at motor level. Electrodes placed horizontally on the submental region 1 cm lateral to the midline above hyoid bone and the other 1 cm latero-posterior to the midline just below the hyoid bone. Physical therapy program (45 min, range of oromotor and oral stimulation exercises—unclear if these were individualised) 3 times a week for 6 weeks (plus 3 times daily independently) Control group 2: Standard medical treatment only	Primary outcomes: Swallowing variables (OTT, hyoid elevation, laryngeal elevation, oesophageal sphincter opening, aspiration/penetration) as per VFSS. Secondary outcome: N/R	OTT significantly improved in Treatment group 1 post intervention (*p* = 0.001) Significantly higher number of patients in Treatment group 1 who had lower aspiration/penetration rate (*p* = 0.008), improved hyoid elevation (*p* = 0.002) and laryngeal elevation (*p* = 0.001). No differences seen in oesophageal sphincter opening
Guillén-Solà, et al. [31]	Assess the therapeutic effectiveness of NMES and inspiratory and expiratory muscle training (IEMT) in dysphagic subacute stroke patients, compared to standard swallow therapy (DT)	Procedure: DT, IEMT, NMES Delivery and dosage: DT: 5 days a week. Self-management education, individualised oral exercises, compensatory techniques based on VFSS IEMT: 5 sets of 10 respirations twice a day 5 days per week for 3 weeks. Loads were set to 30% of max insp and exp pressures, increased weekly by 10 cm H_2_O Sham IEMT: same frequency, but with set workloads of 10 cm H_2_O NMES: 40 min a day 5 days per week for 3 weeks at 80 Hz on suprahyoid muscles	Primary outcomes: Max inspiratory + expiratory muscle function (MicroRPM), dysphagia severity (VFSS, PAS), respiratory complications. Secondary outcomes: Swallowing parameter changes as per voice changes, coughing, desaturation (>3%), piecemeal deglutition, oropharyngeal residue (V-VST), FOIS, DOSS. (Not reported in study) Assessed at baseline, 3 weeks post (by V-VST), and 3 months post intervention (VFSS).	Respiratory muscle strength: Positive treatment effect in the IEMT group at 3 weeks only Dysphagia severity: No significant differences for PAS scores between groups Improved safety at 3 weeks for IEMT and NMES; improved efficacy at 3 months for IEMT Respiratory complications: No adverse effects reported 15.5% with lung infection (4 in DT, 3 in NMES, 2 in IEMT) throughout the follow-up period
Heijnen, et al., [32]	To compare the effects of traditional speech therapy exercises to those combined with NMES on motor or sensory level on dysphagia and quality of life of patients with Parkinson’s Disease	Procedure: NMES with VitalStim protocol DT included oromotor exercises, swallow manoeuvres and strategies 13–15 sessions, 30 min each, on five consecutive days a week over 3–5 weeks Treatment group 1 DT Treatment group 2 DT NMES to the suprahyoid muscle Stimulation to motor level Treatment group 3 DT NMES to the suprahyoid muscle Stimulation to sensory level	Primary outcomes: Health related quality of life (SWAL-QOL; MDADI). Secondary outcomes: Dysphagia severity (single-item Dysphagia Severity Scale)	No significant differences between groups Significant improvement (*p* < 0.001) on Dysphagia Severity Scale for all groups. Restricted positive effects on QOL
Huang, et al. [33]	To compare functional dysphagia recovery in acute stroke patients using traditional dysphagia therapy, NMES or the two combined	Procedure: NMES, DT 10 sessions, 3 x a week, 60 min each VitalStim protocol with electrode placement in a vertical line with one above and one below the thyroid notch Intensity level individual—determined once patient felt a tingling sensation and a muscle contraction DT: oromotor exercises, compensatory techniques, thermal-tactile stimulation, swallow manoeuvres individualised as per VFSS Treatment group 1 DT Treatment group 2 NMES Treatment group 3 DT + NMES DT performed during NMES	Primary outcomes: FOIS, PAS, FDS as per VFSS before and after treatment. Secondary outcome: N/R	No significant differences between groups post therapy in FOIS or PAS scales For FDS, 2 of 4 scales were significantly different (improved) in Treatment group 3 (*p* = 0.03) compared with Treatment groups 1 and 2 Significant differences in FOIS before and after therapy in all 3 groups (*p* = 0.03; *p* = 0.01; *p* = 0.005) Significant differences in PAS before and after therapy in treatment groups 1 and 3 (*p* = 0.04 for both)
Huh, et al. [34]	To investigate the effect of different electrode placement in NMES in poststroke dysphagia rehabilitation	Procedure: NMES (VitalStim protocol with stimulation at motor level) + effortful swallow five 20 min sessions weekly for four weeks Treatment group 1 NMES with horizontal electrode placement One pair of electrodes on the suprahyoid muscles, second pair on the infrahyoid muscles Treatment group 2 NMES with horizontal + vertical electrode placement One pair horizontally on the suprahyoid muscles, second pair vertically on the infrahyoid muscles Treatment group 3 NMES with vertical electrode placement along the midline from hyoid bone down to below the thyroid cartilage	Primary outcomes: VFSS performed at baseline and post treatment. FDS—both oral phase (FDS-O) and pharyngeal phase (FDS-*p*) separately, also DOSS ^b^ Secondary outcome: N/R	Treatment Group 1 scores for FDS and FDS-*p* were significantly higher than those in Groups 2 and 3 No statistically significant differences between groups in FDS-O or DOSS scores post treatment All groups showed significant improvement in FDS (*p* < 0.01) and DOSS (*p* < 0.01) scores post treatment Horizontal electrode placement on the suprahyoid and infrahyoid muscles was found to be more beneficial for dysphagia recovery
Jing, et al. [35]	To investigate the effect of NMES on post stroke dysphagia	Procedure: NMES, DT Treatment for consecutive 10 days Both groups received general medical treatment, and DT (exercises for tongue, mouth and facial muscle function; sensory stimulation; vocal cord; chewing training; therapeutic feeding) Treatment group 1: VitalStim as per protocol, though intensity of 6 to 21 mV. Electrode placement selected based on the patient’s dysphagia presentation: (a) vertical distribution on each side of the midline with lowest electrode just above the superior thyroid notch (b) 1st channel horizontally and close to the surface of the hyoid bone with 2nd channel horizontally along the midline just below the superior thyroid notch (c) 1st channel vertically below the chin and 2nd channel along the buccal branch of the facial nerve Treatment group 2: Intensity of swallow rehabilitation exercises NR	Primary outcomes: Swallow efficacy, swallow function scores, laryngeal elevation, severity of aspiration, amount of food intake, residue scores. All based on Rattans dysphagia classification criteria. Secondary outcome: N/R	Efficacy, laryngeal elevation and severity of aspiration in the treatment group were significantly better post treatment than in the control group (*p* < 0.05) Swallow function scores improved in both groups, but more pronounced in the treatment group (*p* < 0.05) Amount of food intake or residue scores were not significantly different between the two groups
Langmore, et al. [36]	To investigate the efficacy of NMES combined with swallow exercises in improving dysphagia post radiotherapy for head & neck cancer	Procedure: NMES (BMR NeuroTech 2000 default settings with minor alterations) or sham Electrodes placed supra-hyoid region. Home-based protocol, performed 2 x day, 6 days/week, for 12 weeks (3 training sessions to ensure competence). 16–20 min per session DT during treatment sessions: 10 x super-supraglottic, 10x Mendelsohn, 10 x effortful swallows Sham/Treatment group: Sham-NMES delivered via a similar device with wires inside the equipment disconnected. Same session structure and intensity of treatment	Primary outcome: Swallowing function as measured by PAS on VFSS. Secondary outcomes: OPSE, hyoid excursion, diet measured by the PSS, and quality of life as measured by HNCI. Assessments were performed prior to, midway through (week 7) and at the end of the treatment (week 13).	Mean PAS: greater improvement in the sham group (*p* = 0.027). No other outcomes showed a significant difference between the two groups. Treatment group: No significant change in PAS score Significant decrease in the anterior hyoid excursion (*p* = 0.038） No sigificant differences in OPSE Significant improvement in diet (total PSS score, *p* < 0.001) and HNCI quality of life scores for eating (*p* < 0.001) and speech (*p* = 0.016) Sham/Treatment group: Significant improvement in PAS score (*p* < 0.001) No significant differences in OPSE Significant improvement in diet (total PSS score, *p* < 0.046) and HNCI quality of life scores for eating (*p* = 0.003) and speech (*p* = 0.001)
Lee, et al. [37]	To compare early NMES combined with DT versus DT only on dysphagia outcomes in acute/subacute ischaemic stroke patients with moderate to severe dysphagia	Procedure: DT in both groups included thermal-tactile stimulation with any combination of lingual strengthening exercises, laryngeal adduction-elevation exercises, and swallow manoeuvres by SLP 60 min/day for 15 days Treatment group 1: NMES simultaneously with DT for first 30 min max tolerable intensity (120% of the mean threshold value) on both suprahyoid muscles. Pulse rate of 80 Hz with 700 microsec duration 30 min a day, 5 days per week for 3 weeks Treatment group 2:DT only, as per above	Primary outcome: FOIS ^b^ as per VFSS at 3, 6, and 12 weeks post treatment. Secondary outcome: N/R	FOIS: Both groups showed significant improvement in FOIS 3 & 6 weeks post treatment Treatment group 1 showed significant improvement at 12 weeks FOIS: significantly greater improvement in treatment group 1 (at all timepoints) when compared to the treatment group 2 (*p* < 0.05)
Li, et al. [38]	To assess whether adding NMES to the conventional swallow therapy improves post-stroke dysphagia	Procedure: NMES with VitalStim, electrical current level approx 7 mA. No other stimulation data given Electrodes placed supra-hyoid (top electrodes) and infra-hyoid (bottom electrodes) 4 weeks of treatment, 1 h sessions, 5 x week DT included basic training of organs related to food intake and swallowing (no further details given) and direct food intake training (intake environment, body posture for swallowing and removal of residue) Treatment group 1: NMES + DT Treatment group 2: NMES Treatment group 3: DT	Primary outcomes: VAS to compare the differences of muscle pain pre and post treatment; SSA, sEMG, OTT, PTT, LCD and Standardised swallowing PAS were measured using VFSS. Secondary outcome: N/R	SSA scores significantly higher in Treatment Group 1 (*p* < 0.01) compared to groups 2 and 3 Significant decrease in OTT and PTT for liquid and paste bolus (*p* < 0.05 for both) in Treatment Group 1 compared to Groups 2 and 3 No change in LCD Significant increase in max amplitude of sEMG signal in Treatment Group 1 compared to Groups 2 and 3 No significant changes between Groups 2 and 3 SSA scores and maximum amplitude of sEMG signal increased significantly within each group
Maeda, et al. [39]	To investigate the effect of transcutaneous electrical sensory stimulation (TESS) without muscle contraction in patients undergoing dysphagia rehabilitation	Procedure: Sensory stimulation or sham, plus usual treatment (details NR) for both groups using Gentle Stim (J Craft, Osaka, Japan). Beat frequency of 50 Hz, other details NR 2 pairs of electrodes (frequencies of 2000 and 2050 Hz). Anterior electrodes placed at the edge of the thyroid cartilage and the posterior electrodes 4 cm from the ipsilateral electrode along the mandible 15 min of twice daily intervention, 5 days per week for 2 weeks Treatment group: Stimulation intensity set at 3.0 mA Sham group: Stimulation intensity set at 0.1 mA	Primary outcomes: Cough latency time against 1% citric acid mist. Secondary outcomes: FOIS, oral nutritional intake outcomes measured at study entry, and after the 2nd and 3rd week following treatment initiation	No statistically significant differences were found between or within groups Changes in cough latency time and FOIS scores indicated better outcomes in the TESS group, based on substantial effect sizes
Meng, et al. [40]	To assess the effectiveness of surface NMES with various electrode placements on patients with post-stroke dysphagia	Procedure: All groups received DT 30 min per treatment, 5 x week. 10 sessions NMES with VitalStim (Treatment Groups 1 and 2) for 30 min prior to daily DT NMES as per VitalStim with minimum degree of stimulation to induce visible muscle contraction DT combination of therapeutic exercises, compensatory manoeuvres and diet texture modifications. It remains unclear whether these were standard or individual according to VFSS results Treatment Group 1: Electrode placement: 1 pair of electrodes on the surface of both sides of suprahyoid, and another pair on surface of upper and lower edge of thyroid cartilage Treatment Group 2: Electrode placement: 2 pairs of electrodes on the surface of suprahyoid (geniohyoid + mylohyoid) Treatment Group 3: DT	Primary outcomes: VFSS pre and post treatment. Hyoid excursion, DOSS ^b^, WST and RSST. Secondary outcome: N/R	WST, RSST and DOSS scores improved significantly more for Treatment Groups 1 and 2 compared to Control Group (*p* < 0.05) Differences not statistically different between treatment group 1 and 2 WST, RSST and DOSS improved significantly in all groups comparing pre-and post-treatment (*p* < 0.05) VFSS: only increased anterior movement of the hyoid improved statistically significantly and only in Treatment Group 2, pre-post treatment (*p* = 0.006)
Nam, et al. [41]	To assess the effect of repeated sessions of NMES with two different electrode placements on dysphagia following brain injury	Procedure: Hyolaryngeal electrical stimulation 10–15 sessions over 2–3 weeks, one session daily for 30 min Both groups also received simultaneous DT—individual swallow manoeuvres based on VFSS findings Treatment Group 1: Electrode placement on the suprahyoid muscles Stimulation delivered using Stimplus (Cuber-Medic Corp., Iksan, South Korea) Pulse frequency 60 Hz with 500 ms pulse interval Treatment Group 2: Electrode placement on the suprahyoid and infrahyoid muscles Stimulation delivered using VitalStim (Chattanooga Group, Hixson, TN, USA) as per VitalStim protocol	Primary outcomes: Motion analysis of the hyolaryngeal excursion according to VFSS conducted before and after the treatment Secondary outcome: N/R	No significant differences between groups Treatment Group 1 showed a significant increase in the maximal anterior excursion of the hyoid (*p* = 0.008) and the anterior excursion velocity (*p* = 0.017) Treatment Group 2 showed a significant increase in the maximal superior excursion and the maximal absolute excursion distance of laryngeal elevation (*p* = 0.013 for both)
Oh, et al. [42]	To identify the effects of NMES with two different electrode placements on post-stroke dysphagia	Procedure: NMES with VitalStim, as per protocol 30 min/day, 5 days/week for 4 weeks Effortful swallow performed during stimulation Both groups received DT—unclear if this was individualised Treatment group 1: Electrode placement on the suprahyoid muscles Treatment group 2: Electrode placement on the infrahyoid muscles DT included thermal-tactile stimulation, various exercises, manoeuvres, modified food material, viscosity and posture	Primary outcomes: VDS, PAS ^b^ and FOIS Secondary outcome: N/R	PAS improved more in Treatment Group 1 compared to Group 2 (*p* = 0.036). No other significant differences between groups. Treatment Group 1: Significant improvement in VDS (*p* = 0.001), PAS (*p* = 0.002) and FOIS (*p* = 0.014) Treatment Group 2: Significant improvement in VDS (*p* = 0.002), PAS (*p* = 0.045) and FOIS (*p* = 0.026) *NB*. Data as per Table 2 (inconsistencies between text vs table)
Ortega, et al. [43]	To evaluate the effectiveness of two different sensory stimulation treatments on oropharyngeal dysphagia in the elderly	Procedure: Sensory stimulation for 2 weeks Treatment group 1: Chemical sensory stimulation with a natural TRPV1 (capsaicin) agonist solution. Treatment was taken by patients three times a day before each meal and 5 days per week (Mon-Fri) for 2 weeks Treatment group 2: Electrical stimulation using the thyroid position (VitalStim, as per protocol) Intensity 75% of the motor threshold Once a day 5 days per week (Mon-Fri) for 2 weeks	Primary outcome: VFSS measurements, PAS (measured before and 5 days after the treatment) Secondary outcomes: EAT-10, V-VST,	No between group differences reported Treatment group 1: Significant improvement in EAT-10 scores (*p* = 0.016), and safety based on VFSS (*p* = 0.019) Treatment group 2: Significant improvement in safety (*p* = 0.019) and penetrations (*p* = 0.044) based on VFSS
Park, et al. [44]	To determine whether effortful swallow training combined with surface electrical stimulation as a form of resistance training has an effect on post-stroke dysphagia	Procedure: NMES with VitalStim, 2 sets of electrodes placed on infrahyoid muscles (working against resistance) 3 sets of 20 min exercise/week over 4 weeks Treatment group 1: Effortful swallow + NMES (treatment level) NMES as per VitalStim protocol, intensity increased until muscle activation Treatment group 2: Effortful swallow + NMES (non-treatment level)	Primary outcome: Hyolaryngeal excursion (max anterior hyoid displacement, max vertical hyoid displacement), maximum vertical laryngeal displacement, UES opening (width), PAS (as per VFSS), pre and post treatment. Secondary outcome: N/R	Between groups significant difference post treatment NR Treatment group 1: Significant increase in laryngeal elevation (*p* > 0.05). NS increase in vertical hyoid motion and UES opening Treatment group 2: NS difference between any pre-post measures
Park, et al. [45]	To investigate the effects of effortful swallowing combined with NMES on hyoid bone movement and swallowing function in stroke patients	Procedure: NMES (VitalStim, as per protocol), electrodes placed on infrahyoid muscles (targeting sternohyoid muscle, working against resistance) Delivery and dosage: 30 min per session, 5 sessions a week for 6 weeks. Treatment group 1: Effortful swallow + NMES (treatment level) NMES intensity gradually increased until grabbing sensation Treatment group 2: Effortful swallow + NMES (placebo level) Sensory NMES intensity gradually increased until tingling sensation	Primary outcomes: As per VDS pre and post treatment (6 weeks). Kinematics of the hyoid bone (analysed with Image J Program); swallow function (as per VDS and PAS ^b^); VDS measures: Oral phase (lip closure, bolus formation, mastication, apraxia, tongue to palate contact, premature bolus loss and OTT); Pharyngeal phase (pharyngeal triggering, vallecular residues, pyriform sinus resides, laryngeal elevation, pharyngeal wall coating, pharyngeal transit time and aspiration). Secondary outcome: N/R	Significantly greater improvements shown by the treatment group versus the placebo group Treatment group 1: Significant improvements post treatment for VDS total score (*p* < 0.01), VDS pharyngeal phase (*p* < 0.01), vertical and horizontal hyoid bone displacement (*p* < 0.01) and PAS (*p* < 0.01). Improvement for VDS oral phase = NS. Treatment group 2: Vertical and anterior hyoid elevatio*n* = NS (*p* = 0.06, *p* = 0.09 respectively) Significant improvement in total VDS score (*p* = 0.02) and oral phase (0.04). Pharyngeal phase improvement = NS (*p* = 0.07) PAS improvement = NS (*p* = 0.06)
Park, et al. [46]	To identify the effect of effortful swallowing combined with neuromuscular electrical stimulation NMES in treating dysphagia in Parkinson’s disease	Procedure: NMES (VitalStim) 5 days/week, for 4 weeks, 30 min each session During stimulation, patient produced effortful swallow (saliva) Infrahyoid electrode placement After NMES, patients received 30 min DT (orofacial exercises, thermal tactile stimulation and manoeuvres) Treatment group 1: NMES + effortful swallow Treatment group 2: Sensory NMES + effortful swallow Stimulation applied at 1.0 mA, no increase	Primary outcome: Kinematics of the hyoid bone (analysed with Image J Program); swallow function (as per VDS and PAS ^b^) Secondary outcomes: VDS measures: Oral phase (lip closure, bolus formation, mastication, apraxia, tongue to palate contact, premature bolus loss and OTT); Pharyngeal phase (pharyngeal triggering, vallecular residues, pyriform sinus resides, laryngeal elevation, pharyngeal wall coating, pharyngeal transit time and aspiration)	Hyoid bone movement: Significant improvement (*p* < 0.05) with vertical and horizontal movement versus sensory NMES PAS: Significant improvement (*p* < 0.05) as compared with sensory NMES No significant difference between groups with any VDS parameters
Permsirivanich, et al. [47]	To compare the treatment outcomes between dysphagia rehabilitation exercises and NMES in post-stroke dysphagia	Procedure: Treatment administered 5 days a week (Mon-Fri) for 4 weeks Both groups received diet modifications and oromotor exercises if weakness present Treatment group 1: Swallowing rehabilitation exercises Individual based on VFSS findings, may have included thermal stimulation, head & neck positioning and swallow manoeuvres Treatment group 2: NMES using VitalStim, as per protocol Vertical electrode placement—from 1mm above the thyroid notch down past the thyroid notch Treatment level at grabbing sensation 60 min per session	Primary outcomes: Changes in FOIS ^b^, complications related to treatment and number of therapy sessions. VFSS only performed pre-treatment. Secondary outcome: N/R	Improvement in FOIS was significantly greater for Treatment group 2 (*p* < 0.001) No complications related to treatment, no significant difference in the number of sessions received
Ryu, et al. [48]	To evaluate the effect of NMES on dysphagia following treatment for head and neck cancer	Procedure: 30 min of NMES (VitalStim) or transcutaneous electrical stimulation (TENS) Followed by 30 min DT for (oral motor exercises, pharyngeal swallowing exercises, use of compensatory strategies during meals, thermal/tactile stimulation, Mendelsohn manoeuvre and diet-texture modifications) 5 days per week for 2 weeks Treatment group 1 : Electrodes placed horizontally immediately above the thyroid notch (Chanel 1), and parallel below notch (Chanel 2) NMES as per VitalStim protocol Sham/Treatment group 2: Sham stimulation using low intensity TENS	Primary outcome measures: FDS, CDS, ASHA-NOMS and MDADI Secondary outcome: N/R	Significant difference (*p*= 0.04) between the treatment and sham group post intervention for FDS only No significant difference between groups for CDS, ASHA-NOMS nor MDADI
Simonelli, et al. [49]	To investigate the effect of laryngopharyngeal NMES on poststroke dysphagia	Procedure: NMES and/or DT. Treatment 30 min twice daily, 5 days/week for 8 weeks, by SLTs Treatment group 1: NMES (VitalStim) plus DT. Electrode placement 3B (two electrodes were placed just at or above the level of the thyroid notch over the thyrohyoid muscle) Treatment group 2: DT included oral-facial, lingual, laryngeal adduction-elevation exercises, effortful swallow maneuver, Mendelsohn maneuver, Masako maneuver, Shaker exercises and thermal stimulation plus compensatory strategies	Primary outcome: FOIS, PAS ^b^, the Pooling score and the presence of oropharyngeal secretion as per FEES. Secondary outcomes: Diet taken by mouth; the need for postural compensations and the duration of the dysphagia training.	Significant difference between groups for FOIS (*p* = 0.15), PAS (*p* = 0.003) and presence of oropharyngeal secretions (*p* = 0.048), with significantly greater improvements in the NMES group. No difference in pooling score. Significant difference between groups for all secondary outcomes, with significant improvements for the NMES group (*p* < 0.01)
Song, et al. [50]	To investigate the effects of NMES and oral sensorimotor treatment (OST) on dysphagia in children with CP	Procedure: OST followed by NMES (20min) with thickened fluid, delivered by occupational therapist Electrodes placed approximating suprahyoid muscles (Chanel 1) and infrahyoid muscles (Chanel 2) 2 x week for 8 weeks Treatment group 1: OST = sensory stimulation to cheeks, chin, lips, tongue and palate using fingers, vibrator, ice-stick 20 min NMES (Simplus DP 200) 3–5 mA, 80 Hz of 300 milliseconds with 1-s interval Sham/Treatment group: OST + sham-NMES (device not switched on)	Primary outcomes: (1) BASOFF: jaw closure, lip closure over a spoon, tongue control, lip closure while swallowing, swallowing food without excess loss, chewing food (tongue/jaw control), sipping liquids, swallowing liquids without excess loss, and swallowing food without coughing; (2) ASHA-NOMS. Secondary outcome: N/R	Significant difference (*p* < 0.05) between groups for total BASOFF scores post treatment Significant improvements for the treatment group 1 including lip closure while swallowing, swallowing food without excess loss, sipping liquid, swallowing liquid without excess loss, swallowing without cough, and total score No significant changes between or within groups for ASHA-NOMS scores
Sproson, et al. [51]	To investigate the efficacy of the Ampcare Effective Swallowing Protocol (ESP), combining NMES with swallow-strengthening exercises, compared with usual care in the treatment of dysphagia post-stroke	Procedure: NMES to suprahyoid muscles via AmpCare ESP Treatment group 1: 30 min, 5 days/week, 4 weeks NMES pulse rate 30Hz with three sets of 10 min exercises (a) chin to chest against resistance + effortful swallow, (b) chin to chest + Mendelshohn + effortful swallow, (c) chin to chest against resistance + jaw opening-closing + effortful swallow Usual Care Group 2: Usual care varied from periodic reviews primarily focusing on posture and diet modification to weekly visits with home-practise regimes. These regimes included exercises and postural adaptations based on VFSS findings	Primary outcomes: (1) FOIS and PAS ^b^ immediately post treatment as per VFSS; (2) FOIS, PAS and SWAL-QOL 1 month follow-up. Secondary outcome: N/R	No significant difference between groups for any of the outcome measures Descriptive statistics reported FOIS: 62% of NMES patients improved (versus 50% of standard care) PAS: Variable results reported SWAL-QOL: 83% of NMES patients improved (versus 38% of standard care)
Terré, et al. [52]	To evaluate the effectiveness of neuromuscular electrical stimulation NMES treatment in patients with oropharyngeal dysphagia secondary to acquired brain injury	Procedure: NMES (VitalStim), or sham, + traditional dysphagia therapy, 60 min, 5 days/week for 4 weeks Treatment group 1: Stimulation as per VitalStim protocol Electrode placement: submental/suprahyoid region and infra hyoid region Plus DT (individualised from VFSS): diet modification, supraglottic, Mendelsohn manoeuvre, oromotor exercises Sham/Treatment group 2: Sham NMES + DT Electrode placement = chin region and lateral to thyroid with minimal stimulus (2.5 mA) to top electrode Sham stimulation with DT	Primary outcome: FOIS Secondary outcomes: VFSS parameters, pharyngo-esophageal manometry Assessed at 1 month (immediately post therapy) and at 3 months.	Significant difference between groups at 1 month (greater improvement with treatment group). No significant difference between groups at 3 months. Secondary outcomes: VFSS: Statistically fewer patients from treatment group aspirated (nectar and pudding) at 1 month. No significant difference at 3 months. Pharyngo-esophageal manometry: difference between groups not reported
Umay, et al. [53]	To evaluate the effects of sensory electrical stimulation (SES) to bilateral masseter muscles in early stroke patients with dysphagia	Procedure: Sensory level electrical stimulation (Intelect Advanced) with galvanic stimulation to bilateral masseter muscles for 60 min, 5 days/week, for 4 weeks Treatment group 1: Sensory stimulation established when patient reported tingling sensation. Electrical current level 4–6 mA Combined with DT: dietary modification, and oromotor exercises, though not during stimulation Sham/Treatment group 2: Electrode placement without stimulation DT as per above	Primary outcomes: Bedside dysphagia score (from water swallow test, pulse oximetry), total dysphagia score, MASA, NEDS. Secondary outcome: N/R	Significant difference between groups post treatment = NR Pre-post treatment changes (improvements) were significantly greater in the treatment group with bedside dysphagia score (*p* = 0.015), total dysphagia score (*p* = 0.001), MASA (*p* = 0.004) and NEDS (*p* = 0.001)
Umay, et al. [54]	To investigate the effects of sensory-level electrical stimulation NMES treatment applied to bilateral masseter muscles at the lowest current level combined with conventional dysphagia rehabilitation in children with CP who had any oropharyngeal dysphagia symptoms	Procedure: Sensory-level NMES (with Intelect Advanced) 30 min/day, 5 days/week for 4 weeks DT given separately, 30 min/day, 5 days/week for 4 weeks Treatment group 1: Sensory-level ES + DT Sensory-level ES to bilateral masseter muscles, at lowest current level where child showed signs of discomfort (sensory threshold). No oropharyngeal exercises or swallow training performed at the same time. DT by rehabilitation specialist: daily care for oral hygiene, thermal (cold) and tactile Stimulation, head and trunk positioning and dietary modification. Oral motor exercises included for children who could participate. Sham/Treatment group 2: Sham ES + DT Sham ES = same electrode placement, no stimulus DT as per above	Primary outcome: Ped EAT-10, FEES; Secondary outcomes: Clinical Feeding Evaluation.	Significantly greater improvement for treatment group versus sham with both Ped EAT-10 and FEES. (Though difference between groups post therapy not reported). Secondary outcomes: Statistically greater changes (effect size) for clinical feeding parameters: drooling, tongue movements, chewing and feeding duration for the treatment group versus sham
Xia, et al. [55]	To investigate the effects of VitalStim therapy coupled with conventional swallowing training on recovery of post-stroke dysphagia	Treatment group 1: Standard swallow therapy (DT). Schedule not reported Direct and indirect OD training related to food intake and swallowing, body posture and removal of pharyngeal food residue Treatment group 2: NMES (VitalStim), 30 min, 2 x day. 5 days/week for 4 weeks Treatment group 3: DT + VitalStim Schedule not reported	Primary outcome: Dysphagia Rating Scale ^b^ (as per VFSS); Secondary outcomes: Maximum amplitude of surface electromyography (sEMG) signals of hyoid muscles; SWAL-QOL.	Primary outcomes: All 3 groups significantly improved post treatment. Significant greater improvement (*p* < 0.01) for group 3 (DT + VitalStim) versus other 2 groups (DT only group and VitalStim only group). Secondary outcomes. SWAL-QOL and sEMG signals significantly increased in all groups. Significant difference between DT + VitalStim (greater improvement) versus DT group and VitalStim group.
Zeng, et al. [56]	To observe the improvement of swallow function and negative affect disorders in patients with cerebral infarction and dysphagia by NMES	Procedure: NMES and/or swallow training NMES via YS1002T Glossopharyngeal Nerve and Muscle Electrical Stimulator (Changzhou Yasi Medical Instruments Co) Stimulation pulse width of 800 ms, intensity 28 mA Swallow training included: massage to cheeks, tongue, retropharyngeal wall, pharyngopalatine arch and lips with frozen cotton swabs or fingers soaked in ice water. Followed by an empty swallow. Treatment group 1: 5. Swallow training only 6. Dose/schedule not reported Treatment group 2: 7. Swallow training + NMES 8. NMES for 20 min period in intervals of 3 s, daily for 12 days. After a 2 day break, NMES for another 12 days.	Primary outcome: Swallow function as per Kubota water-drinking test; Secondary outcomes: Negative affect disorders as per Hamilton anxiety scale and depression scale test.	Primary outcomes: Both groups improved swallow function post treatment, significantly greater improvements (*p* = 0.035) for group 2 (swallow training + NMES) Secondary outcomes: Anxiety and depression subscales and scores improved significantly only in treatment group 2. Significant difference between the groups post treatment for anxiety scales (*p* = 0.001) and depression scales (0.033).
Zhang, et al. [57]	To evaluate and compare the effects of NMES acting on the sensory input versus motor muscle in treating patients with dysphagia with medullary infarction	Procedure: Electrical stimulation via vocaSTIM-master + 2 surface electrodes, placed submentally. Pulse width = 100 ms; frequency = 120 Hz. 20 min, 2 x day, 5 days/week for 4 weeks. Treatment group 1: Standard swallowing therapy (DT): postural adjustment, diet modification, thermal-tactile stimulation, oromotor exercises, swallow manoeuvres Dosage and schedule not reported Treatment group 2, DT + sensory NMES: Stimulation intensity 0–15 mA, increasing to ‘sensory input’. Treatment group 3, DT + motor NMES: Stimulation intensity 0–60 mA, increasing to maximal tolerable level.	Primary outcomes: WST, FOIS, SWAL-QOL, SSA. Secondary outcome: N/R	All treatment groups improved significantly (*p* < 0.01) pre-post across all outcome measures Significantly greater treatment effect was noted for DT + sensory NMES compared to other two treatment groups, across all measures (*p* = 0.01–0.04) Significantly greater treatment effect was noted for DT + motor NMES compared to DT only
**Pharyngeal Electrical Stimulation (PES)—*n* = *8***				
Bath, et al. [58]	Assess the efficacy of PES in treating subacute poststroke dysphagia	Procedure: PES (Phagenyx) catheter + standard stroke care 3 days, 10 min/day Standard stroke care included thrombolysis; rehabilitation; antihypertensive agents; if indicated, oral antithrombotic, lipid-lowering agents and carotid endarterectomy (ischemic stroke patients) Treatment group: 10 min stimulation, PES (mA) at 75% of difference between max tolerance level and threshold level Sham: Phagenyx catheter inserted, no stimulation after threshold and max tolerance level obtained	Primary outcome: PAS ^b^ (via VFSS), assessed at 2 and 12 weeks post treatment. 3–7 bolus per VFSS. Secondary outcome: At 2, 6 and 12 weeks = DSRS, function (Barthel Index), dependency (modified Rankin Scale), impairment (NIHSS), quality of life (EQ-5D), nutritional measures and serious adverse events (chest infections, pneumonia, death).	No significant difference (*p* = 0.60) in dysphagia improvement between treatment and sham group Treatment group: PAS mea*n* = 3.7 (2.0) Sham group: PAS mea*n* = 3.6 (1.9) Authors conclude: PES is safe but did not improve dysphagia. May be impacted by PES ‘under-treatment’/suboptimal dose
Dziewas, et al. [59]	Assess the safety and efficacy of PES in accelerating dysphagia rehabilitation and enabling decannulation of tracheostomised stroke patients	Procedure: PES (Phagenyx) 10 min/day, 3 consecutive days Treatment group: 10 min stimulation calculated using patient´s perceptual threshold and max tolerated threshold Sham group: Phagenyx catheter inserted, no stimulation after threshold and max tolerance obtained Open label PES group: Following post-treatment assessment, all patients who had not improved were offered active PES treatment as per above schedule	Primary outcome: Readiness for decannulation 24–72 h after treatment (determined by FEES protocol) Secondary outcomes: delayed improvement in Open label group; recannulations (between 2–30 days post decannulation/discharge); DSRS; FOIS; stroke severity as per modified Rankin Scale and NIHSS; LOS, SLT plan, number and type of adverse events.	Primary outcomes: 17/35 patients (49%) ready for decannulation versus sham 3/34 (9%) patients. Significant difference (*p* < 0.001) between groups Secondary outcomes: Open-label PES (a) Retreated group = 4/15 (27%) ready for decannulation (b) Sham/delayed treatment group = 16/30 (53%) ready for decannulation. No significant differences between groups.
Essa, et al. [60]	Assess if The Brain Derived Neurotrophic Factor (BDNF) genotype can influence swallowing recovery post PES in stroke patients	Procedure: PES Once a day for 10 min on 3 consecutive days Treatment group PES—0.2 ms pulses, 280 V with 5Hz frequency at 75% max tolerated intensity Sham group Sham PES	Primary outcome: DSRS. Assessed at baseline, 2 weeks and 3 months post treatment. Secondary outcome: N/R	No between group statistics reported In the treatment group, the genotype Met carriers of the BDNF gene had significant improvement in DSRS by 3 months post intervention (*p* = 0.009), when compared to those homozygous for the Val allele No significant improvement in the Sham group Data support the notion that the presence of the Met allele might be a predictor of improved long-term outcomes for dysphagia after PES
Fraser, et al. [61]	To assess the effect of PES on swallow function in hemispheric stroke patients	Procedure: PES Single session of 10 min 5 Hz with max tolerated intensity for treatment group Sham group received no stimulation	Primary outcomes: PTT, swallowing response time, PAS Secondary outcome: N/R	Between group statistics = NR Treatment group showed a significant pre-post reduction in pharyngeal transit time, swallowing response time and PAS (all *p* < 0.01) No difference in pre-post (change) outcomes for the sham group
Jayasekeran, et al. [62]	To examine the role of PES in expediting human swallowing recovery after experimental (virtual) and actual (stroke) lesions	Agent: PES Protocol 1—active or sham PES with virtual lesion Patients their own controls. The two studies (active or sham) took place at least 1 week apart. Protocol 2—PES with varying treatment intensity (times/day) and dose (total number of days) Group 0—no stimulation Group 3—once/day for 3 days Group 5—once/day for 5 days Group 9—3 times/day for 3 days Group 15—3 times/ day for 5 days Protocol 3—active or sham PES with acute stroke. Once daily on three consecutive days.	Primary outcomes: Protocol 1 Cortical excitability, swallow timeliness Protocol 2 PAS ^b^ Protocol 3 PAS ^b^, swallow timing, DSRS, LOS at hospital, Barthel Index. For protocols 2 and 3, VFSS conducted before treatment, and again weeks later. Secondary outcome: N/R	Protocol 1 Active PES abolished the effects of virtual lesion by reversing the direction of excitability. Active PES reversed the direction of cortical excitability in both hemispheres (*p* = 0.42). Active PES abolished the behavioural effects of the virtual lesion (*p* = 0.02), increasing the number of correctly timed swallows by 65% Protocol 2 Intensity (times/day): Compared to control, once/day stimulation (groups 3 and 5) produced the greatest reduction in aspiration (*p* = 0.04) Dose: Compared to control, total of 3 days of stimulation (groups 3 and 9) showed the greatest reduction in aspiration scores (*p* = 0.038) Protocol 3 Reduction of PAS post intervention for the active PES group compared to sham = NS (*p* = 0.49) No significant changes in swallow timing for either group Significantly reduced DSRS in the PES group (*p* = 0.04) NS shorter stay in hospital for the PES group (*p* = 0.38)
Restivo, et al. [63]	To investigate whether intraluminal electrical pharyngeal stimulation facilitates swallowing recovery in dysphagic multiple sclerosis (MS) patients	Procedure: PES (bipolar platinum pharyngeal ring electrodes built into 3 mm-diameter intraluminal catheter) using constant/current electrical simulator (DS7) Stimulation 10 min, 5 consecutive days Treatment group: 5 Hz pharyngeal stimulation (mA calculated using sensory threshold and pain thresholds, mea*n* = 14.2 ± 0.6 mA) Sham: Same catheter, no stimulation	Primary outcome: PAS via VFSS at pre-treatment (T0), immediately after treatment (T1), after two (T2), and four (T3) weeks of PES. Secondary outcomes: sEMG measure of: (1) duration of laryngeal excursion; (2) duration of the sEMG activity of suprahyoid/submental muscles; (3) duration of the inhibition of the CP muscle; and (4) interval between onset of suprahyoid/submental muscles and onset of laryngeal elevation.	Significant difference between treatment and sham group immediately and 4 week post treatment, for PAS (*p* < 0.0001) and all secondary measures (*p* < 0.0001) Treatment group improved significantly across all measures, sham group did not
Suntrup, et al. [64]	To assess the effectiveness of PES on swallowing function of severely dysphagic tracheostomised patients	Procedure: PES (Phagenyx) catheter system and base station, stimuli of 0.2 ms pulse duration at a frequency of 5 Hz with 280 V Stimulation 10 min, 5 consecutive days Treatment group: Stimuli of 0.2 ms pulse duration at a frequency of 5 Hz with 280 V Sham: Same catheter, no stimulation Another treatment session was offered to participants who were not eligible for tracheostomy decannulation post the first treatment session.	Primary outcome: Eligibility for decannulation Secondary outcomes:FOIS at discharge; mRS; LOS in ICU and hospital; time from stimulation to discharge.	75% of the treatment group participants were able to be decannulated post Tx compared to 20% of sham group (*p* < 0.01) No significant differences in the secondary outcomes between groups A further 71.4% of participants were able to be decannulated post second round of treatment
Vasant, et al. [65]	To assess the effectiveness of PES on swallowing in poststroke dysphagia, with clinical effects in longer-term follow-up	Procedure: PES (Gaeltec catheter) inserted nasally or orally (patient preference) 10 min stimulation for 3 consecutive days Treatment group: PES: stimuli delivered (0.2 ms pulses, maximum 280 V) at defined optimal parameters (5 Hz frequency and an intensity [current] 75% of maximum patient toleration Additional DT as determined by SLP assessment (details not supplied) Sham: PES catheter insitu, no stimulation. DT by SLP.	Primary outcome: DSRS at 2 weeks post treatment. Secondary outcomes: DSRS at 3 months, feeding method, PAS ^b^ (as per MBS/FEES), number of adverse events (chest infections, death).	Primary outcome: significant difference between groups NR Treatment group effects (DSRS measures) were noted at 2 weeks and 3 months post treatment, though not significant (*p* = 0.26 and 0.97 respectively) No significant difference reported between groups for most secondary outcomes
**Combined Neurostimulation Interventions—*n* = *4***				
Cabib et al. [66]	To investigate the effect of repetitive transcranial magnetic stimulation (rTMS) of the primary sensory cortex (A), oral capsaicin (B) and intra-pharyngeal electrical stimulation (IPES; C) on post stroke dysphagia	Procedure: All patients received both treatment and sham, cross over active/sham in visits 1 week apart (randomised). Assessment occurred immediately prior to treatment and within 2 h post treatment. Treatment group 1: rTMS (Magstim rapid stimulator) Stimulation (90% of threshold) bilaterally to motor hotspots for pharyngeal cortices 5 Hz train of 50 pulses for 10 sec x 5 (total 250 pulses), 10 sec between trains Sham = coil tilted 90 degrees. Treatment group 2: Capsaicin stimulus (10−5 M) or placebo (potassium sorbate) were administered once in a 100 mL solution Treatment group 3: PES via two-ring electrode naso-pharyngeal catheter (Gaeltec Ltd) 10 min stimulation at 75% tolerance threshold (0.2 ms of duration) and 5 Hz Sham = 30 seconds of above stimulation then no stimulation	Primary outcomes: Effect size pre-post treatment for neurophysiological variables (pharyngeal and thenar RMT and MEP). Secondary outcomes: Effects on the biomechanics of swallow (PAS ^b^, impaired efficiency + more) VFSS before and after treatment	Between group differences (post treatment) not reported Primary outcomes: No significant differences in pre-post pharyngeal RMTs with any of the active or sham conditions Combined analysis (interventions grouped together) showed significantly shorter latency times, increased amplitude, and area of the thenar MEP in the contralesional hemisphere Secondary outcomes: (VFSS) No significant change/difference in effect size across any of the treatment or sham groups
Lim, et al. [67]	To investigate the effect of low-frequency repetitive transcranial magnetic stimulation (rTMS) and neuromuscular electrical stimulation (NMES) on post-stroke dysphagia	Procedure: DT: oropharyngeal muscle-strengthening, exercise for range of motion of the neck/tongue, thermal tactile stimulation, Mendelson manoeuvre, and food intake training for 4 weeks Treatment group 1: DT 4 weeks Intensity NR Treatment group 2: DT + rTMS via Magstim 200 (Magstim, Whiteland, UK) Stimulation to pharyngeal motor cortex, contralateral hemisphere 1 Hz stimulation, 100% intensity of resting motor threshold 20 min/day, (total 1200 pulses a day), 5 x week for 2 weeks Treatment group 3: DT + NMES (Vitalstim) 300 ms, 80 Hz (100 ms in interstimulus intervals). Intensity between 7–9 mA, depending on patient compliance Stimulation to supra and infra hyoid region 30 min/day, 5 days/week, 2 weeks	Primary outcomes: VFSS baseline, 2 weeks + 4 weeks post treatment (for semi-solids and liquids) FDS, PTT, PAS. Secondary outcome: N/R	Difference between groups post treatment = NR FDS outcome: For semi-solids all groups improved, no significant difference in pre-post change, between groups For liquids, the rTMS and NMES improved significantly compared to DT, 2 weeks post treatment (*p* = 0.016 and *p* < 0.001, respectively) No significant difference in the change from baseline to the 4th week evaluation among groups (*p* = 0.233) PAS outcome: For semi-solids all groups improved, no significant difference in pre-post PAS change, between groups For liquids, the rTMS and NMES improved significantly compared to DT, 2 weeks post treatment (*p* = 0.011 and *p* = 0.014, respectively) No significant difference in the change from baseline to the 4th week evaluation among groups (*p* = 0.540)
Michou, et al. [68]	To compare the effects of a single application of one of three neurostimulation techniques (PES, paired stimulation, rTMS) on swallow safety and neurophysiological mechanisms in chronic post-stroke dysphagia	Procedure: Single application of neurostimulation All patients received real and sham treatment in randomised order on two different days Treatment group 1: PES Frequency of 5 Hz for 10 min. Intensity set at 75% of the difference between perception and tolerance thresholds Treatment group 2: Paired associative stimulation: Pairing a pharyngeal electrical stimulus (0.2 ms pulse) with a single TMS pulse over the pharyngeal MI at MT intensity plus 20% of the stimulator output. The 2 pulses were delivered repeatedly every 20 s with an inter-stimulus interval of 100 ms for 10 min. Treatment group 3: rTMS Stimuli to pharyngeal motor cortex with the TMS coil. Frequency of 5 Hz, intensity 90% of resting thenar motor threshold in train of 250 pulses, in 5 blocks of 50 with 10 s between-blocks pause.	Primary Outcome: VFSS before and after treatment (PAS ^b^) Secondary outcomes: Percentage change in cortical excitability; OTT, pharyngeal response time, PTT, airway closure time and upper oesophageal opening time as per VFSS	Treatment group 1 (PES): significant excitability increase immediately post-Tx in the unaffected hemisphere (real vs sham *p* = 0.043) and in the affected hemisphere 30min post-Tx (real vs sham *p* = 0.04) With Paired Stimulation, cortical excitability increased 30 min post-Tx in the unaffected side (*p* = 0.043) compared to sham, and immediately post-Tx in the affected hemisphere following contralateral Paired stimulation (*p* = 0.027) Treatment group 2 (paired neurostimulation): an overall increase in corticobulbar excitability in the unaffected hemisphere (*p* = 0.005) with an associated 15% reduction in aspiration (*p* = 0.005) when compared to sham Pharyngeal response time was significantly shorter post treatment with real stimulation compared to sham (*p* = 0.007) Treatment group 3 (rTMS): an increase in excitability in the unaffected hemisphere, but no significant difference compared to sham. No change in the affected hemisphere. Corticobulbar excitability of pharyngeal motor cortex was beneficially modulated by PES, Paired Stimulation and to a lesser extent by rTMS
Zhang, et al. [69]	To determine whether repetitive transcranial magnetic stimulation (rTMS) combined with neuromuscular electrical stimulation (NMES) effectively ameliorates dysphagia and how rTMS protocols (bilateral vs. unilateral) combined with NMES can be optimized	Procedure: 9. 10 rTMS (sham or real) and 10 NMES sessions Mon-Fri during 2 weeks 10. NMES: 30 min once daily using a battery powered handheld device (HL-08178B; Changsha Huali Biotechnology Co., Ltd., Changsha, China), vertical placement of electrodes. Pulse width of 700 ms, frequency 30–80 Hz, current intensity 7–10 mA. 11. rTMS delivered by figure-of-eight coil (CCY-IV; YIRUIDE Inc., Wuhan, China) during NMES with a sequence of HF-rTMS over the affected hemisphere followed by LF-rTMS over the unaffected hemisphere. 12. HF-rTMS parameterss: 10 Hz, 3 s stimulation, 27 s interval, 15 min, 900 pulses, and 110% intensity of resting motor threshold (rMT) at the hot spot. 13. LF-rTMS parameters: 1 Hz, total of 15 min, 900 pulses, and 80% intensity of rMT at the hot spot. Treatment group 1: Sham rTMS + NMES 10 Hz sham rTMS delivered to the hot spot for the mylohyoid muscle at the ipsilesional hemisphere followed by 1 Hz sham rTMS over the corresponding position of the contralesional hemisphere. Delivered using a vertical coil tilt, generating the same noise as real rTMS without cortical stimulation. Treatment group 2: Ipsilateral rTMS + NMES 10 Hz real rTMS was delivered to the hot spot for the mylohyoid muscle at the ipsilesional hemisphere followed by 1 Hz sham rTMS over the corresponding position of the contralesional hemisphere. Treatment group 3: Contralateral rTMS + NMES 10 Hz sham rTMS was delivered to the hot spot for the mylohyoid muscle at the ipsilesional hemisphere followed by 1-Hz real rTMS over the corresponding position of the contralesional hemisphere Treatment group 4: Bilateral rTMS + NMES 10 Hz real rTMS was delivered to the hot spot for the mylohyoid muscle at the ipsilesional hemisphere followed by 1-Hz real rTMS over the corresponding position of the contralesional hemisphere	Primary outcome: cortical excitability (amplitude of the motor evoked potential) Secondary outcomes: SSA and DD.	Compared with group 2 or 3 in the affected hemisphere, group 4 displayed a significantly greater % change (*p*.0.017 and *p*.0.024, respectively). All groups displayed significant improvements in SSA and DD scores after treatment and at 1-month follow-up. The % change in cortical excitability increased over time in either the affected or unaffected hemisphere in treatment groups 1, 2 and 4 (*p* < 0.05). In Group 3, the % change in cortical excitability in the unaffected hemisphere significantly decreased after the stimulation course (*p* < 0.05). Change in SSA and DD scores in group 4 was markedly higher than that in the other three groups at the end of stimulation (*p*.0.02, *p*.0.03, and *p*.0.005) and still higher than that in group 1 at the 1-month follow-up (*p*.0.01).

^a^ NMES is at motor stimulation level unless explicitly mentioned. ^b^ Data included in meta-analyses. Notes. ASHA-NOMS—American speech-language-hearing association national outcome measurement system; BASOFF— behavioural assessment scale of oral functions in feeding; BI—Barthel index; CDS—clinical dysphagia scale; CNS—central nervous system; CP—cerebral palsy; CT—computed tomography; CVA—cerebrovascular accident; DD—degree of dysphagia; A-DHI—Arabic dysphagia handicap index; DOSS—dysphagia outcome and severity scale; DSRS—dysphagia severity rating scale; DT—dysphagia therapy; EAT-10—eating assessment tool-10; EES— electrokinesiographic/electromyographic study of swallowing; EQ-5D—European Quality of Life Five Dimension; FDS—functional dysphagia scale; FOIS—functional oral intake scale; FEDSS—fiberoptic endoscopic dysphagia severity scale; FEES—fiberoptic endoscopic evaluation of swallowing; HNCI—head neck cancer inventory; IADL—instrumental activities of daily living; ICH—intracranial haemorrhage; ICU—intensive care unit; LPM—laryngeal-pharyngeal mechanogram; MASA—Mann assessment of swallowing ability; MDADI—M.D. Anderson dysphagia inventory; LCD—laryngeal closure duration; LOS—length of stay; MBS—modified barium swallow; MBSImp—modified barium swallow impairment profile; MEG—magnetoencephalography; MMSE—mini-mental state exam; MEP—motor evoked potentials; MRI—magnetic resonance imaging; mRS—modified rankin scale; MS—multiple sclerosis; NEDS—neurological examination dysphagia score; NIHSS—national institutes of health stroke scale; NIHSS—National Institutes of Health Stroke Scale; NMES—neuromuscular electrical stimulation; NS—not significant; OD—oropharyngeal dysphagia; OPSE—oropharyngeal swallow efficiency; OST—oral sensorimotor treatment; OTT—oral transit time; PAS—penetration–aspiration score; PED EAT-10 pediatric eating assessment tool-10;PES—pharyngeal electrical stimulation; PESO— pharyngoesophageal segment opening; PPS—performance status scale; PTT—pharyngeal transit time; RMT— resting motor threshold; RSST—repetitive saliva swallowing test; rTMS—repetitive transcranial magnetic stimulation; SAH—subarachnoid haemorrhage; SAPP—swallowing activity and participation profile; SDQ—swallowing disturbance questionnaire; sEMG—surface electromyography; SFS—swallow function score; SHEMG—electromyographic activity of the submental/suprahyoid muscles complex; SI—similarity index; SLT—speech and language therapist; SSA—standardised swallowing assessment; SWAL-QOL—swallowing quality of life; TBI—traumatic brain injury; tDCS—transcranial direct current stimulation; TOR-BSST—Toronto bedside swallowing screening test; UES—upper esophageal sphincter; UPDRS—unified Parkinson’s disease rating scale; VAS—visual analogue scale; VFSS—videofluoroscopic swallowing study; VVS-T—volume viscosity swallow test; WST—water swallow test.

**Table 4 jcm-11-00776-t004:** Between subgroup meta-analyses for NMES and pharyngeal electrical stimulation (PES) comparing intervention groups of included studies.

Neurostimulation	Subgroup	Hedges’ *g*	Lower Limit CI	Upper Limit CI	*Z*-Value	*p*-Value
NMES	Diagnostic groups					
	Aged dysphagia [>65 yrs] (*n* = 1)	0.291	−0.299	0.881	0.966	0.334
	Cerebral palsy (children) (N = 2)	0.264	−0.088	0.616	1.470	0.142
	Head and neck cancer (*n* = 2)	0.281	−0.610	1.172	0.618	0.536
	Parkinson’s disease (*n* = 2)	0.000	−0.359	0.359	0.000	1.000
	Stroke (*n* = 9)	0.433	0.105	0.760	2.589	0.010 *
	Intervention types					
	NMES (*n* = 2)	0.134	−0.247	0.515	0.688	0.492
	NMES + DT (*n* = 7)	0.648	0.398	0.897	5.086	<0.001 *
	Time between pre-post (days)					
	14 (*n* = 1)	−0.099	−0.888	0.690	−0.246	0.806
	21 (*n* = 1)	1.013	0.466	1.559	3.631	<0.001 *
	28 (*n* = 6)	0.342	−0.062	0.746	1.657	0.098
	56 (*n* = 1)	0.751	0.040	1.462	2.069	0.039 *
	Outcome measures					
	DOSS (*n* = 2)	0.188	−0.407	0.784	0.621	0.535
	FOIS (*n* = 2)	0.805	0.268	1.343	2.937	0.003 *
	PAS (*n* = 2)	0.235	−0.799	1.269	0.446	0.656
	VFSS-scale 1 (*n* = 1)	−0.099	−0.888	0.690	−0.246	0.806
	VFSS-scale 2 (*n* = 2)	0.611	−0.193	1.415	1.489	0.137
	Total stimulation time (min)					
	Low [< 500 min] (N = 4)	0.317	−0.304	0.938	0.999	0.318
	Medium [500–100 min] (N = 1)	−0.099	−0.888	0.690	−0.246	0.806
	High [>100 min] (N = 4)	0.607	0.176	1.038	2.761	0.006 *
	Electrodes configuration					
	Infrahyoid (N = 3)	0.771	0.041	1.501	2.069	0.039 *
	Mixed (patient-dependent) (N = 2)	0.617	−0.195	1.429	1.489	0.137
	Suprahyoid and infrahyoid (N = 2)	0.056	−0.0544	0.655	0.182	0.856
	Suprahyoid (N = 2)	−0.100	−0.694	0.493	−0.331	0.740
	Pulse duration (μs)					
	300 (N = 1)	0.751	0.040	1.462	2.069	0.039 *
	350 (N = 3)	0.084	−0.391	0.559	0.348	0.728
	700 (N = 4)	0.680	0.227	1.133	2.944	0.003 *
	*Pulse rates (Hz)*					
	30 (N = 1)	−0.304	−1.082	0.473	−0.768	0.433
	80 (N = 8)	0.519	0.202	0.836	3.206	0.001 *
PES	Total stimulation time (min)					
	10 (N = 2)	0.300	−0.325	0.925	0.940	0.347
	30 (N = 3)	0.053	0.245	0.351	0.348	0.728

Note. * Significant.

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
