# Peer review of "Neurostimulation in People with Oropharyngeal Dysphagia: A Systematic Review and Meta-Analyses of Randomised Controlled Trials—Part I: Pharyngeal and Neuromuscular Electrical Stimulation"

_jcm, 2022, doi:10.3390/jcm11030776_

Round 1

Reviewer 1 Report

This meta-analysis evaluated the effects of NMES and PES in patients with oropharyngeal dysphagia. Overall 30 RCTs on NMES and 8 RCTs on PES were included to the analyses and between group analyses showed small effect sizes of NMES and no significant effects for PES.

Major comment

  1. Table 2 and Table 3 are missing in the manuscript. Since both the characteristics of participants and the outcome measures are essential in evaluating the methodological soundness, reviewing is impossible without detailed descriptions of them.
  2. In the abstract, the authors stated that 30 studies on NMES were included to the analyses, however, in the result section, only 18 studies were synthesized for pre-post analysis and 9 studies for between group analysis for NMES instead of 30. Similarly, only 5 studies were synthesized for PES instead of 9. The authors have stated they decided to exclude studies to reduce heterogeneity for comparability. However, this may be misleading in that the readers might think that those numbers in the abstract are came out from more studies.
  3. One of the significance of this study is the fact that this study did not exclude populations based on medical diagnosis as stated in the introduction as well as the discussion section. However, the main outcomes of this study were actually derived from studies that only included stroke patients. Again, it seems to be an inevitable choice, however, the authors should this clearly in the limitation section because it can be misleading.

Reviewer 2 Report

Dear authors, I would like to congratulate you on the preparation of this systematic review.
The preparation and processing will certainly lead to a significant improvement in the understanding of the therapy of swallowing dissorders.
I would ask you to revise a few points in your manuscript:
1)Figure 1 Some informations are still missing between the section identification and screening
you write n=8059 records identified and n=908 recrds removed before screening (707+201) n=6946 records are screened,

According to my calculation n=205 studies are missing.
Please specify this flowchart.

2)
You correctly describe that the study situation is very inhomogeneous.
Due to the inhomogeneity, the conclusion that NMES should be preferred compared to PES would be a bit far reaching.
A weakening of these statements has to be discussed.

Round 2

Reviewer 1 Report

  1. The revised manuscript still doesn't have Table 2 and 3. The characteristics of the participants and outcome measures provides key information for interpreting the results, but it does not seem to be clearly described in that respect. While the methodology section is well described, however, in the results section, details are not explained other than that outcome measures varied greatly across the studies. It needs to be described in detail in the text which outcome measure(s) was(were) used in the meta-analyses, in addition to being presented in the Tables.

  1. The systematic review includes 30 studies on NMES and 8 studies on PES; and to reduce the heterogeneity and enhance comparability, meta-analyses were conducted for 9 studies and 5 studies. This point should be clearly stated. The revised abstract is better than the original abstract, but I think this should be made clearer. I understand it is not easy to summarize your study just in 200 words, however, I think you should prioritize which information to include according to its importance. In my opinion, how many studies were included to the systematic review and the meta-analyses is an essential component in abstract.

In the same context, if the meta-analysis was conducted only including stroke patients, it should be stated in the conclusion (i.e., in the conclusion section, the first sentence could be changed to “Meta-analyses for RCTs in NMES found a significant, large pre-post intervention effect size and significant, small post-intervention between-group effect size in favor of NMES in stroke patients.”). Because the authors mentioned that one of the advantages of this study is not to exclude patients by medical diagnosis, it is easy for the readers to naturally assume that this conclusion was drawn from a study involving all patients with oropharyngeal dysphagia.

I’m sorry to keep making similar comments, but I hope the authors kindly understand because I believe that in the scientific paper, the less ambiguity that could lead to unnecessary misunderstanding is the better.
